

# High Resolution Model Intercomparison Project phase 2 (HighResMIP2) towards CMIP7

Malcolm J. Roberts[1], Kevin A. Reed[2], Qing Bao[3], Joseph J. Barsugli[4,5], Suzana J. Camargo[6], Louis-Philippe Caron[7], Ping

Chang[8], Cheng-Ta Chen[9], Hannah M. Christensen[10], Gokhan Danabasoglu[11], Ivy Frenger[12], Neven S. Fučkar[13,14], Shabeh ul

Hasson[15], Helene T. Hewitt[1], Huanping Huang[16,17], Daehyun Kim[18], Chihiro Kodama[19], Michael Lai[1], Lai-Yung Ruby

Leung[20], Ryo Mizuta[21], Paulo Nobre[22], Pablo Ortega[23], Dominique Paquin[7], Christopher D. Roberts[24], Enrico Scoccimarro[25],

Jon Seddon[1], Anne Marie Treguier[26,27], Chia-Ying Tu[28], Paul A. Ullrich[29], Pier Luigi Vidale[30], Michael F. Wehner[31], Colin

M. Zarzycki[32], Bosong Zhang[33], Wei Zhang[34], Ming Zhao[35]

[1] Met Office, FitzRoy Road, Exeter EX1 3PB, UK
[2] School of Marine and Atmospheric Sciences, Stony Brook University, USA
[3] Institute of Atmospheric Physics, Chinese Academy of Sciences, Beijing
[4] CIRES, University of Colorado, Boulder, CO, USA
[5] NOAA Physical Sciences Laboratory, Boulder, CO, USA
[6] Lamont-Doherty Earth Observatory, Columbia University, Palisades, NY, USA
[7] Ouranos, Montréal, Quebec, Canada
[8] Department of Oceanography, Texas A&M University, College Station, TX 77840 USA
[9] Department of Earth Sciences, National Taiwan Normal University, Taipei
[10] Department of Physics, University of Oxford, UK
[11] US National Science Foundation National Center for Atmospheric Research, Boulder, CO, USA
[12] GEOMAR Helmholtz Centre for Ocean Research Kiel, Kiel, Germany
[13] Environmental Change Institute, University of Oxford, Oxford, UK
[14] Earth Sciences Department, Barcelona, Spain
[15] HAREME Lab, Institute of Geography, CEN, Universität Hamburg, Germany
[16] Department of Geography and Anthropology, Louisiana State University, Baton Rouge, LA, USA
[17] Climate and Ecosystem Sciences Division, Lawrence Berkeley National Lab, Berkeley, CA, USA
[18] School of Earth and Environmental Sciences, Seoul National University, Seoul, South Korea
[19] Japan Agency for Marine-Earth Science and Technology, Yokohama, Japan
[20] Pacific Northwest National Laboratory, USA
[21] Meteorological Research Institute, Japan
[22] National Institute for Space Research, São José dos Campos, SP, Brazil
[23] Barcelona Supercomputing Center, Spain
[24] ECMWF, Shinfield Park, Reading, UK
[25] CMCC Foundation - Euro-Mediterranean Center on Climate Change, Italy
[26] Laboratoire d'Océanographie Physique et Spatiale (LOPS), University of Brest, Brest, France
[27] CNRS, IRD, Ifremer, IUEM, Brest, France
[28] Research Center for Environmental Changes, Academia Sinica, Taipei
[29] Lawrence Livermore National Laboratory, USA
[30] National Centre for Atmospheric Science, Dept. of Meteorology, University of Reading, Reading, UK
[31] Lawrence Berkeley National Laboratory, USA
[32] Department of Meteorology and Atmospheric Science, Pennsylvania State University, University Park, PA USA



[33] Program in Atmospheric and Oceanic Sciences, Princeton University, Princeton, USA

[34] Department of Plants, Soils and Climate, Utah State University, USA

[35] Geophysical Fluid Dynamics Laboratory, National Oceanic and Atmospheric Administration (NOAA), Princeton, USA

*Correspondence to*: Malcolm J. Roberts (malcolm.roberts@metoffice.gov.uk)

**Abstract.** Robust projections and predictions of climate variability and change, particularly at regional scales, rely on the driving processes being represented with fidelity in model simulations. Consequently, the role of enhanced horizontal

resolution in improved process representation in all components of the climate system continues to be of great interest. Recent simulations suggest both the possibility of significant changes in large-scale aspects of the ocean and atmospheric circulations and the regional responses to climate change, as well as improvements in representations of small-scale processes and extremes, when resolution is enhanced.

The first phase of HighResMIP (HighResMIP1) was successful in producing a baseline multi-model assessment of global

simulations with model grid spacings of 25-50 km in the atmosphere and 10-25 km in the ocean, a significant increase when compared to models with standard resolutions of order 1-degree typically used as part of the Coupled Model Intercomparison Project (CMIP) experiments. In addition to over 250 peer-reviewed manuscripts using the published HighResMIP1 datasets, the results were widely cited in the Intergovernmental Panel on Climate Change report and were the basis for a variety of derived datasets, including tracked cyclones (both tropical and extratropical), river discharge, storm

surge, and others that were used for impact studies. There were also suggestions from the few ocean eddy-rich coupled simulations that aspects of climate variability and change might be significantly influenced by improved process representation in such models.

The compromises that HighResMIP1 made should now be revisited, given the recent major advances in modelling and computing resources. Aspects that will be reconsidered include experimental design and simulation length, complexity, and

resolution. In addition, larger ensemble sizes and a wider range of future scenarios would enhance the applicability of HighResMIP.

Therefore, we propose an updated HighResMIP2 to improve and extend the previous work, to address new science questions, and to further advance our understanding of the role of horizontal resolution (and hence process representation) in state-of-the-art climate simulations. With further increases in high-performance computing resources and modelling

advances along with the ability to take full advantage of these computational resources, an enhanced investigation of the drivers and consequences of variability and change in both large- and synoptic-scale weather and climate is now made possible. With the arrival of global cloud-resolving models (currently run for relatively short timescales), there is also an opportunity to improve links between such models and more traditional CMIP models, with HighResMIP providing a bridge to link understanding between these domains. HighResMIP also aims to link to other CMIP projects and international efforts

such as the World Climate Research Program lighthouse activities and various Digital Twin initiatives, as well as having the potential to be used as training and validation data for the fast evolving Machine Learning climate models.



## 1 Introduction

Understanding the connections between large-scale climate change and local impacts remains a significant challenge. One approach used by the scientific community to advance our capabilities in this area is through the development of higher horizontal resolution (i.e., reduced grid spacing) General Circulation Models (GCMs) (e.g. Satoh et al., 2008; Small et al., 2014; Wehner et al., 2014; Zhao et al., 2009) to better represent key components of the Earth system (e.g., Athanasiadis et al., 2022; Camargo and Wing, 2016; Chen and Lin, 2011; Shaevitz et al., 2014). Over the last five decades, the horizontal resolutions that are used for the World Climate Research Programme (WCRP) Coupled Model Intercomparison Project (CMIP6; Eyring et al., 2016) have continually improved. More recently, efforts to simulate the global atmosphere and ocean at grid spacings of less than 50 km have become more commonplace in the community (e.g., Caldwell et al., 2019; Chang et al., 2020; Giorgetta et al., 2018; Roberts et al., 2019; Scoccimarro et al., 2022) with advancements in numerical models and high-performance computing. Storm-resolving models with grid spacings of 10 km and less (e.g., Stevens et al., 2019) allow us to further expand our scientific understanding of climate processes and impacts at regional and local scales.

The High Resolution Model Intercomparison Project phase 1 (hereafter HighResMIP1; Haarsma et al., 2016) was a new project within CMIP Phase 6. Its main focus was to develop a better understanding of the role of increased horizontal resolution in climate simulations via a simple experimental design which would be affordable for many modelling groups. It proposed using around 25 km grid spacing in atmosphere (and ocean) as the baseline high-resolution model configuration, with a comparable lower resolution counterpart at ~100 km (consistent with an equivalent CMIP6 model configuration). As of early 2024, 17 modelling groups using 40 different models (including both lower and higher resolution versions) had contributed data to the Earth System Grid Federation (ESGF CMIP6 HighResMIP Data Holdings, 2024) linked to HighResMIP1, of which 21 models were coupled atmosphere-ocean-sea-ice (IPCC, 2021: Annex II: Models, Table AII.6). More than 250 peer-reviewed papers using HighResMIP data have been published to date (Fig. 1 displays a word cloud compiled from these papers), the HighResMIP1 paper has been cited over 580 times, and there were more than 150 direct references to HighResMIP in the Intergovernmental Panel on Climate Change (IPCC) Sixth Assessment Report of Working Group I (IPCC, 2021) putting it in the top three most directly referenced MIPs. Given the broad impacts of the HighResMIP1 outcomes and continuing needs for improved high-resolution climate data in climate change impact assessments, it is important to extend HighResMIP towards CMIP7 and hence continue to provide important insights in time for the next IPCC report.



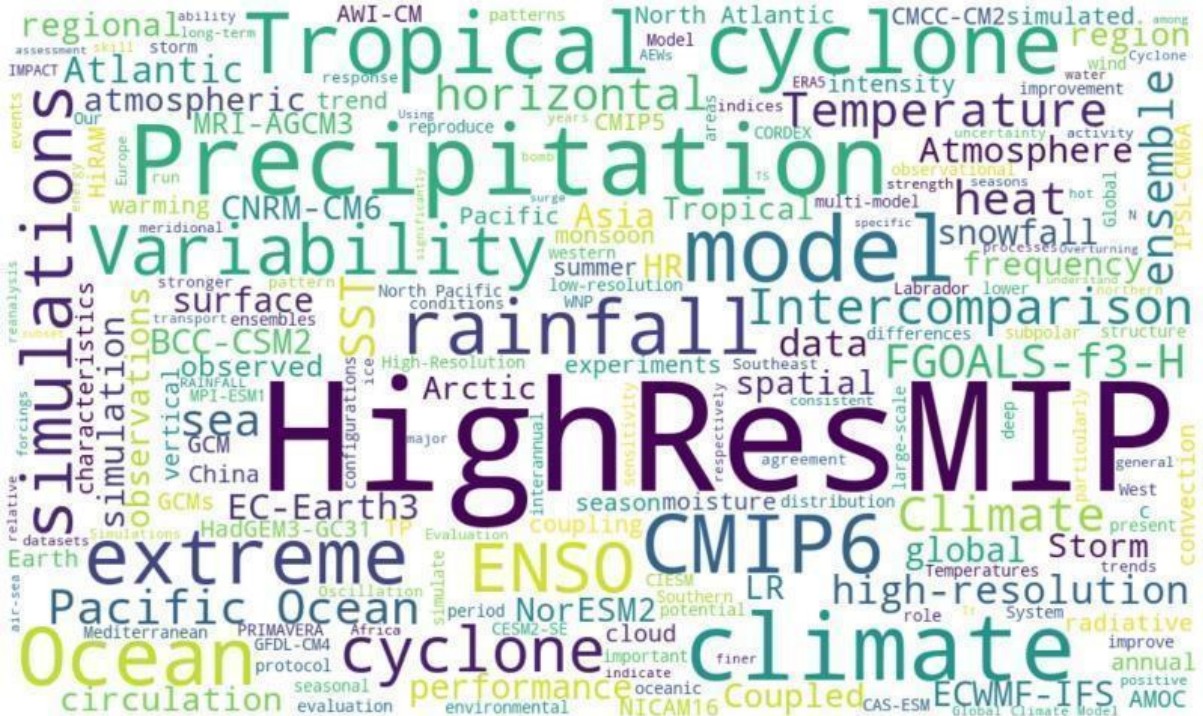

**Figure 1: A word cloud chart of CMIP6 HighResMIP compiled from published papers using its data, which highlights the prominent research areas and applications that the data has been used for.**

The main achievements from HighResMIP1 fall into several categories: quantifying the impacts of model resolution on simulated climate and extreme events (Bador et al., 2020; Gore et al., 2023; Huang et al., 2021; Moon et al., 2022; Roberts et al., 2020a; Scoccimarro et al., 2022); improved understanding of the causes of mean-state model biases (Moreno-Chamarro et al., 2022; Xu et al., 2022); improved process representation, including processes related to topography (Rhoades et al., 2022) and extreme weather (Intergovernmental Panel on Climate Change (IPCC), 2023; Liu et al., 2021; Scoccimarro et al., 2017), and their consequences for simulated climate and climate change; indications that observed large-scale trends (e.g., Eastern Pacific and Southern Ocean cooling) may be captured in high-resolution, but not standard resolution models (Yeager et al., 2023); and associated with better captured trends, generally improved prediction skill (Sobel et al., 2023; Yeager et al., 2023; Zhao and Knutson, 2024). Many groups had previously produced high-resolution simulations individually, but HighResMIP1 provided a protocol for model simulations (both atmosphere-only and coupled) to be consistently performed and compared to each other and available observations, e.g., to understand where and how high-resolution systematically reduce large-scale model biases (e.g., Athanasiadis et al., 2022; Bock et al., 2020; Docquier et al., 2019; Moreno-Chamarro et al., 2022; Scoccimarro et al., 2022; Vannière et al., 2019). Resolutions of 25 km or finer also enabled improved representation of atmospheric extreme processes such as tropical and extratropical cyclones and atmospheric rivers, and ocean mesoscale phenomena and extremes such as marine heatwaves, and hence provide deeper insights into how these



might change in the future (e.g., Bian et al., 2023; Chang et al., 2023; Liu et al., 2021; Priestley and Catto, 2022; Roberts et
al., 2020b; Yamada et al., 2021; Yin et al., 2020; Zhao, 2020, 2022). In addition to these fundamental scientific insights,
HighResMIP outputs were also used to produce datasets useful for impacts and event attribution studies, such as winter
windstorms (Lockwood et al., 2022), synthetic tropical cyclones (Bloemendaal et al., 2022), and global water level change
(Muis et al., 2023).

We also recognise that the HighResMIP1 experimental design had limitations, often due to deliberate choices to make it
more tractable and to enhance process understanding. The relatively short length of simulations made examination of
decadal and longer variability difficult. The strongly parsimonious use of ensembles raised questions about robustly
disentangling externally forced signals from internally-driven variability, including extremes. The suggested use of
simplified aerosol forcing and lack of tuning for these high-resolution models improved the comparability with their lower
resolution counterparts, but was not conducive to producing optimal simulations (as is commonplace for model
development). The short spin-up for the coupled models (based on 1950's conditions) means that the ocean continued to drift
in these HighResMIP1 simulations (Caldwell et al., 2019; Roberts et al., 2019). As a result, it was also difficult to
characterise the models as is conventionally done with CMIP Diagnostic, Evaluation and Characterization of Klima (DECK)
simulations, hence limiting the uptake of the data by the scientific community more broadly. We propose to address some of
these limitations in HighResMIP2, though some compromises will remain to keep the simulations affordable.

In discussions with the scientific community, one important question came up often: Why propose an update to HighResMIP
now? In particular, and given that the CMIP7 experimental design and forcings are not yet available, the community already
has access to existing HighResMIP1 data from simulations that were computationally expensive and take considerable time
to complete. However, the proposed time scale is primarily driven by the HighResMIP community (i.e., the groups with the
computational resources to complete such simulations), with several groups expressing interest in starting new simulations as
soon as 2024. A key reason for this keenness is the increased maturity of models optimised to run at high-resolution, both
scientifically and technically. Although such simulations cannot yet use CMIP7 forcings, knowledge of the proposed
HighResMIP2 design and confidence that new simulations would still be relevant as part of HighResMIP2 analysis, was
very important for these modelling groups (see Table 1). In addition, any simulations following HighResMIP1 protocols can
still have their data published as part of CMIP6. It is noted that for the HighResMIP1 results cited above, the forcings used
were standard CMIP6, and so we have evidence that the details of the forcings (and hence any difference between CMIP6
and CMIP7 forcings) are not central to our high-resolution insights thus far. Another factor that was considered is the time
needed to develop high-resolution configurations, particularly of coupled climate models with eddy-rich ocean components,
which often lags a standard resolution model by years, as well as forward planning to access the large computational
resources necessary. If we want the unique insights from global high-resolution coupled simulations, that are simply not
available from standard-resolution models, to feed into the next IPCC cycle with its fast-approaching deadlines, we need
considerably more lead time for such models. The new HighResMIP2 described here enables this by providing a future path
and giving groups time to plan. We accept that at the time of this paper, HighResMIP2 is effectively bridging across both



CMIP6 and CMIP7, and consequently there may be some confusion in this manuscript when new simulations are described but not all the information is yet available. We have attempted to be as clear as possible, and further details will be documented via standard CMIP protocols at the earliest opportunity.

| Model name | Contact institution | Atmosphere resolution mid-latitude (km) | Ocean resolution mid-latitude (km) | CMIP era (i.e. forcings used) |
|---|---|---|---|---|
| ICON | Max-Plank-Institute for Meteorology (MPI-M) | 10 | 5 | CMIP6 |
| IFS-FESOM2 | Alfred Wegener Institute (AWI) | 9 | Variable, 13-4.5 | CMIP6 |
| IFS-NEMO | Barcelona Supercomputing Center (BSC) | 9 | 8 | CMIP6 |
| HadGEM3-GC5 | UK Met Office | 20 | 8 | CMIP7 |
| CAS-ESM2 | Chinese Academy of Sciences (CAS) | 25 | 10 | CMIP7 |
| FGOALS-f4 | Chinese Academy of Sciences (CAS) | 12.5 | 10 | CMIP7 |
| BCC-CSM3-HR | Chinese Meteorological Agency (CMA) Earth System Modeling and Prediction Centre | 30 | 25 | CMIP7 |
| MRI-ESM3 | Meteorological Research Institute (MRI) | 20 | 10 | CMIP7 |
| NICAM, NICOCO | Japan Agency for Marine-Earth Science and Technology | 14 | 10 | CMIP7 |
| GFDL-C384CM4 | Geophysical Fluid Dynamics Laboratory / NOAA | 25 | 25 with a possibility for 10 | CMIP6/CMIP7 |



**Table 1: Modelling groups proposing to contribute to HighResMIP2 with coupled model simulations (which are more challenging and for which the model diversity was low in HighResMIP1). Many more atmosphere-only simulations are anticipated.**


Given this timing, what can we hope to gain from new simulations? Key limitations of the HighResMIP1 simulation data included: unoptimised high-resolution models (i.e. configured as similar as possible to the standard-resolution counterpart, including simplified aerosol forcing); few ensemble members (generally only one); small diversity of models particularly
with coupled simulations; and resolutions that were generally 25 km or coarser. We have ambition for new simulations to have further enhanced resolution (aiming for around 10 km in atmosphere and ocean, see Table 1), producing larger ensembles and using models that are optimised for high-resolution (scientifically and technically). In addition, we propose new experiments that will help to better characterise the models and build links with other communities. One emerging area where global high-resolution simulation data could play a key role is that of machine learning, which is clearly making huge
advances but is constrained by the quality and quantity of training data. Results from CMIP6, HighResMIP1, and conclusions from IPCC AR6 also suggest that significant scientific uncertainties remain in future projections due to lack of resolution. With such new simulations building on the HighResMIP1 archive, we believe that existing and new science questions proposed in Section 2 can be addressed in a more complete way.

Throughout the paper we refer to "high-resolution" global climate models, which will be defined as models that have smaller
grid spacings than are typical of CMIP6 DECK climate simulations, where the vast majority of models had grid spacings of 100-200 km in the atmosphere and 50-100 km in the ocean (IPCC, 2021, Fig. 1.19). Our current understanding is that such resolutions (henceforth described as standard-resolution) will likely remain typical of CMIP7 (Dunne et al. in prep), given historic rates of increase in model resolution (Hewitt et al., 2022). We will also refer to climate models and Earth System models (ESMs) interchangeably in the text, though we expect that most models in HighResMIP2 will have limited Earth
System complexity (but may, for example, include interactive aerosols which were not recommended in HighResMIP1).

HighResMIP, however, is just one part of community efforts to produce climate information at resolutions beyond the standard CMIP model capabilities, and we aim to enhance our collaborations with these other related initiatives. The DYAMOND project (Stevens et al., 2019) pushed forward the evolution of global storm-resolving climate simulation in a multi-model framework, and this is being expanded with both national efforts and potential international initiatives such as
Earth Virtualization Engines (EVE; Stevens et al., 2024). These global storm-resolving modelling efforts share with HighResMIP a lack of basic characterisation - one example would be metrics of climate sensitivity - and so several new experiments are proposed in HighResMIP2 to improve the links with CMIP models and potentially address this lack of metrics. This work also builds and complements other WCRP activities, including those with similar goals to explore climate change impacts at regional scales using other techniques such as downscaling via the Coordinated Regional Climate
Downscaling Experiment (CORDEX) with often similar resolutions at the regional scale. Furthermore, new international efforts such as the WCRP lighthouse activities, including Digital Earths, Explaining and Predicting Earth System Change,



and Safe Landing Climates activities (Sherwood et al., 2024), as well as the European Destination Earth programme (Hoffmann et al., 2023; Wedi et al., 2022), are well-positioned to make use of HighResMIP2 simulations for exploring extreme events and regional impacts of climate change.

This paper is structured as follows. Section 2 outlines the specific goals of HighResMIP2, as well as the general simulation approach. Section 3 details the experimental design and HighResMIP2 protocols. The data issues are discussed in Section 4, and Section 5 provides an overview of the evaluation and metrics framework to be used for analysis. Finally, Section 6 discusses the main takeaways of the effort.

## 2 Science questions

Three overarching questions frame the scientific scope of HighResMIP2:

Can HighResMIP simulations help to better quantify and even reduce key structural uncertainties of future climate projections discussed in the IPCC AR6 report, and can it contribute to and supplement the CMIP7 AR7 FastTrack science goals, particularly around future weather extremes, the pattern effect (interaction of forcings, feedbacks and natural variability) and tipping points?

What resolution-dependent atmospheric or oceanic processes are missing from standard-resolution CMIP6 models that might explain recent changes in our climate, and produce different future climate projections over the next few decades?

Can we combine information from HighResMIP2 and CMIP7, together with other data sources and processing (e.g., bias correction), to produce more robust plans for future adaptation and climate risk planning?

The main reason that such questions cannot already be answered with existing data is that the HighResMIP1 high-resolution
models were configured to be as similar as possible to their standard-resolution counterparts, and used simplified aerosol forcings. This provided an important baseline for scientific understanding of the role of horizontal resolution on simulation quality, but crucially did not produce optimal simulations of present and future climate at high resolution. In addition, most of the existing simulations do not have eddy-rich ocean resolutions, which limits the processes they are able to explicitly simulate and hence our ability to quantify future uncertainty.

As noted above, CMIP6 HighResMIP1 enabled significant progress in studying the role of horizontal resolution in global climate simulations. Various aspects of global and regional climate have been assessed (HighResMIP publications, 2024), including scale interactions, circulation in the atmosphere and ocean, the role of ocean eddies in upwelling systems, extremes and hydrological cycles, and tropical and extratropical cyclones. The poor representation of these processes due to lack of resolution lead to key uncertainties in future projections (e.g., as described in the IPCC AR6 report, with specific examples
below). The main objective of HighResMIP2 is to build on the HighResMIP1 baseline, to further our understanding of how model resolution affects various aspects of climate simulation, and to assess the implications for future climate projections and climate risk and adaptation planning with optimised high-resolution models. In addition, new simulations are proposed



that will enable new science questions around forcings and feedbacks in the climate system and the role of high-resolution in the pattern effect (i.e. how large-scale sea surface temperature patterns interact with aspects of the climate).

To scope our ambitions, it is useful to consider what model resolutions may be possible for CMIP7 HighResMIP2, given ongoing advancements in models and supercomputing. (Chang et al., 2020, 2023) have demonstrated that it is now possible to produce long CMIP DECK-style simulations (with multiple ensemble members) using coupled model resolutions of 25 km in the atmosphere and 10 km in the ocean component while several groups used similar eddy-rich ocean resolutions (below 10 km) in CMIP6 HighResMIP1 (Caldwell et al., 2019; Chang et al., 2020; Grist et al., 2021; Roberts et al., 2019).

Hence we anticipate more coupled models with such ocean resolution that can more accurately represent the ocean's mesoscale, including eddies, boundary currents, and Southern Ocean processes (see Table 1). We also expect enhancements in atmosphere resolution perhaps to 10 km or finer; this may lead to challenges related to spatial scales at which resolved model physics and dynamics may overlap with convective parameterisation (Hong and Dudhia, 2012), the so-called "grey zone", but has the potential to produce more realistic climate extremes and upscale feedbacks (Scaife et al., 2019). In

addition to resolution alone, many model developments (both scientific and technical) flowing from storm-resolving modelling approaches can also help to optimise model configurations for HighResMIP2.

    It is likely that most, if not all, of HighResMIP2 simulations will only include the physical climate and will lack many aspects of Earth System complexity (for example the carbon cycle and biogeochemistry), given the cost, but such simulations would be warmly welcomed to begin exploring the interactions between resolution and complexity. Other

aspects of complexity, such as representation of ocean tides, ocean waves, ice-ocean interactions including ice shelves and sheets, may be represented in some models, but likely not all.

    We organise the science questions into several broad areas that are enabled by the different simulation types (as detailed in Section 3), atmosphere-only and coupled. We recognise the relative strengths and limitations of prescribed SST or sea ice configurations on the realism of upscale feedbacks, which are complementary to coupled simulations which allow us to

explore a fully interactive system, as well as understand the potential role of SST biases in the coupled simulations.

    Atmosphere-only simulations allow us to ask: given external forcings and prescribed sea surface temperature (SST) and sea-ice, how does the coupled atmosphere-land system respond, via processes and extremes? We can hence ask whether increasing model resolution can lead to fundamentally different insights into large-scale climate variability and future trends, with implications for policy and climate risk. Furthermore, atmosphere-only simulations have the potential to bring new

insights into long-standing questions about model capability and suitability for different applications. Extreme events like precipitation and mesoscale convective systems (MCSs) also require further investigation (e.g., Na et al., 2022; Zhao, 2022). These events have a significant impact on the environment and can cause severe damage to infrastructure and human life. It may also be possible to identify potential future risks associated with specified patterns and levels of warming (Zhao and Knutson, 2024). Atmospheric resolutions of around 10 km are anticipated to offer a much more realistic range of intensities

for extreme events, such as tropical cyclones (TCs; Li et al., 2021) and MCSs. This enhancement could offer new insights into how TC genesis and rapid intensification, as well as MCSs, may change in a warmer climate.



With horizontal resolutions beginning to edge into the "grey zone", aspects of climate variability, such as the Madden–Julian Oscillation, the diurnal cycle, and hot spots with complex terrain like the Pan-Tibetan Plateau, can be addressed (Bao et al., 2020). This opens up new avenues for research and allows us to better understand the underlying mechanisms relevant to

these phenomena and regions. For this reason we propose new short simulations (one year) to incorporate contributions from models with resolutions ranging from standard CMIP through to storm-resolving. These simulations will focus on process-based analysis and linking understanding across different communities.

Coupled simulations will investigate the robustness of projected large-scale changes to the climate as ocean grids are refined down to eddy-rich scales. The ocean mesoscale, which refers to the physical processes that occur on a scale of 10-100 km,

plays a crucial role in key aspects of ocean circulation (e.g., Chang et al., 2020; Chassignet et al., 2020; Hewitt et al., 2022; Small et al., 2014) and shows evidence for substantial changes in a warming world (Beech et al., 2022; Martínez-Moreno et al., 2021); but it is unresolved in standard-resolution CMIP simulations. However, the IPCC AR6 report acknowledges that there is considerable uncertainty in predicting future trends in regions such as the Southern Ocean due to insufficient model resolution (Fox-Kemper et al., 2022). This uncertainty could have significant implications for sea-level rise, as well as the

uptake of heat and carbon (Xu et al., 2023), for example with new observations suggesting an important role for ocean eddies in Antarctic ice shelf melting (Gao et al., 2024). The ocean mesoscale is also important in Eastern Boundary Upwelling Systems, EBUS (Chang et al., 2023), which are nutrient-rich and provide a large proportion of productive biomes. Refined ocean resolution in EBUS leads to narrower and stronger along-shore ocean flow and coastal upwelling, resulting in larger across-shore temperature gradients than in coarse-resolution models (Small et al., 2024). In terms of ocean boundary

currents such as the Gulf Stream, nearly all CMIP6 models (including most of HighResMIP1) have a common bias in which the current separates from the coastline too far north (Chassignet et al., 2020), which can be improved at eddy-rich resolutions with implications for regional climate (Grist et al., 2021; Moreno-Chamarro et al., 2021). Enhancing the representation of ocean transports through narrow straits, such as the Bering Strait, has been demonstrated to have significant upscaling effects on large-scale climate responses, such as Arctic Amplification (Xu et al., 2024). Models with an ocean

resolution of 25 km or finer are beginning to explore these emerging features and their implications for present and future climate (e.g., Chang et al., 2023; Moreno-Chamarro et al., 2021; Rackow et al., 2022). Hence new simulations and more multi-model studies will better constrain, and may be able to reduce, structural uncertainty in future projections in all these regions.

Ongoing analysis of the only existing CMIP6-style multi-centennial simulation with an eddy-rich ocean (Chang et al., 2020)

has demonstrated how such model resolutions can open up new science. In addition to exploring the impacts on mean state and variability, ongoing work (Yeager et al., 2023) indicates that these higher resolution models may capture SST trends of the recent historical period, particularly the relative cooling in the Southern Ocean and tropical Eastern Pacific, which are poorly represented in all CMIP6 models (Seager et al., 2019, 2022). Such results have important implications for climate risk and adaptation planning over the next few decades (Sobel et al., 2023; Zhao and Knutson, 2024), because the state of the

tropical Pacific has implications for teleconnections and regional extremes both at large-scales (e.g., floods and droughts as





seen during El Nino - Southern Oscillation (ENSO) events) and for the distribution of extreme events such as TCs. The model used in Chang et al. (2020) also has improved skill and signal-to-noise properties (Yeager et al. 2023), suggesting that reducing biases in the Southern Ocean is important for better decadal predictions.

In addition to the role that increased resolution might have for the large-scale climate, high-resolution simulations underpin new science endeavours on scale interactions and upscale effects of small-scale processes which are relatively unexplored due to the lack of available simulations. Scoccimarro et al. (2020) show how tropical cyclones can lead to drying of large areas such as the Maritime Continent. For questions such as: How do extreme events impact the main modes of climate variability (e.g., TCs and ENSO modulation), we need models that faithfully represent chains of processes from extreme events back to the large-scale. In a similar vein, Schemm (2023) shows how small-scale diabatic processes can play a key role in addressing long-standing mid-latitude storm track biases. In the ocean, the role of eddies in driving large-scale ocean circulation (for example the Atlantic Meridional Overturning Circulation, AMOC; Hewitt et al., 2020) is a relatively new area for both models and observations to explore (Hirschi et al., 2020). Eddies control the Southern Ocean overturning by compensating wind-driven changes (e.g., Bishop et al., 2016; Farneti et al., 2010, 2015; Gent and Danabasoglu, 2011; Meredith et al., 2012), and this mechanism could evolve in the future as eddy activity increases (Beech et al., 2022). The improvement of the Gulf Stream dynamics in an eddy-rich ocean model has a significant impact on the projected sea level rise on the East Coast of the US (Li et al., 2022).

By using eddy-rich or km-scale models, we might obtain new insights into abrupt changes or tipping points (Armstrong McKay et al., 2022). Abrupt changes can include events like the collapse of the AMOC or Amazon rainforest, or melting of Arctic or Antarctic ice sheets and the consequences for the global climate and extremes. For example, enhanced atmospheric resolution improves the spatio-temporal distribution of rainfall over the Amazon basin due to improved representation of atmospheric dynamics (Monerie et al., 2020). Ocean eddies have also been shown to be important in the development and properties of marine heatwave extremes (Bian et al., 2023).

The IPCC AR6 chapter on extremes described uncertainties in future changes to processes such as TCs, which need both high-resolution and decade-to-century time scale simulations to assess variability and change. It is hoped that HighResMIP2 simulations can help to further address some of these uncertainties. Using a one-year experiment (see Section 3) as a common baseline for models from CMIP6 resolution down to km-scale, we hope to better understand the drivers and processes that govern such extremes (e.g., pre-cursor TC seeds such as MCSs) and hence better constrain likely future changes. As indicated in the Introduction, many of the HighResMIP areas of interest are strongly aligned with national and international projects. We will be working in collaboration with these projects, both via coordinated experiments and data sharing as well as analysis, and we anticipate that this will make HighResMIP2 an invaluable platform for both scientific research and stakeholder engagement.



## 3 Experimental design and protocol

HighResMIP1 focused on the time period 1950-2014 (historical) and 2015-2050 (future) for atmosphere-only and coupled simulations, using only one future scenario (SSP5-8.5) with standard and high-resolution versions of each model, as a
balance between affordability by as many modelling groups as possible, while still retaining multi-decadal timescales to examine model biases, climate variability, and extremes. The standard-resolution versions of the models were meant to have a counterpart in CMIP6 DECK and hence be characterised there. It should be noted that in CMIP6 the historical period was defined as 1850-2014, and future scenarios started from 2015. In CMIP7 we anticipate that the historical period will extend to 2022. Once the CMIP7 forcings (Dunne et al. in prep) are available, HighResMIP2 will provide details of how these
should be implemented in our proposed simulations.

Based on feedback from users and producers of the HighResMIP output data, we propose some key changes to the experimental designs used in HighResMIP1 (see Fig. 2):

1. Extend the coupled future simulation (*highres-future*) to at least 2100, with a change of future scenario away from the high emission SSP5-8.5 to one with lower radiative forcing magnitude, also enabling stronger links with CMIP, CORDEX,
and impacts communities;

2. Encourage groups to produce simulations with models optimised for use at high-resolution (e.g. Hourdin et al., 2017), with methods transparent and documented;

3. Allow use of any aerosol scheme desired (rather than utilising the MacV2-SP aerosol properties scheme (Stevens et al., 2017) as in HighResMIP1) - this is a significant change, to reduce the burden of developing and tuning a separate model
from that used for other modelling activities including CMIP;

4. Reduce the length of the *highresSST-present* experiment (atmosphere-only) to 1980-2022 to focus on a period with better observations, and enable use of a satellite-derived product for SST and sea-ice with a native resolution of ~1/20°, also suitable as a boundary condition for much higher resolution simulations;

5. Remove the *highresSST-future* experiment (atmosphere-only). This attempted to extend *highresSST-present* into the future
(2015-2050) by constructing SST and sea-ice forcings combining observations and model projections. However it was a source of some misinterpretations and highlighted the difficulty in joining the historical forcing to a projected future change in a meaningful way.

We also propose several new experiments, based on our science questions and on an aspiration to better connect with the CMIP DECK simulations:
1. Specific warming level experiments using patterned SST and sea-ice changes, parallel to *highresSST-present* over the 2003-2022 period, as an idealised way to investigate the impact on climate extreme processes;

2. Uniform +4K SST experiment in the atmosphere-only experiment, in order to calculate some overall HighResMIP model metrics and characterisation useful for comparison with CMIP and DYAMOND efforts (this is a standard experiment in the Cloud Feedback Model Intercomparison project, CFMIP);



3. Add a 4xCO2 experiment to the coupled simulations, in order to diagnose Effective Climate Sensitivity (EffCS), though here that measure will be relative to a 1950-control rather than the standard CMIP DECK experiment relative to 1850. There is some evidence that 20-30 years of this simulation may be adequate to estimate the final EffCS (usually after 150 years; Dong et al., 2020);

4. Add a short experiment (~1 year in length) to enhance collaboration across more modelling groups, from standard CMIP-
type models through to km-scale global models, and including numerical weather prediction centres. This atmosphere-only simulation would be in support of the process intercomparison projects (PIPs) advocated by WCRP (World Climate Research Programme, 2022), and have strong links to DYAMOND3 (Takasuka et al., 2024), and enable coordinated detailed analysis (with an expanded diagnostics list).

Figure 2 illustrates the experiments envisioned for HighResMIP2. As with HighResMIP1, we consider several Tiers of
experiments (Table 2), but Tier 1 remains the essential entry card to HighResMIP2, while the others are optional but individually stand as useful community simulations. Figure 3 illustrates how these simulations link to existing simulations in CMIP and other community MIPs.

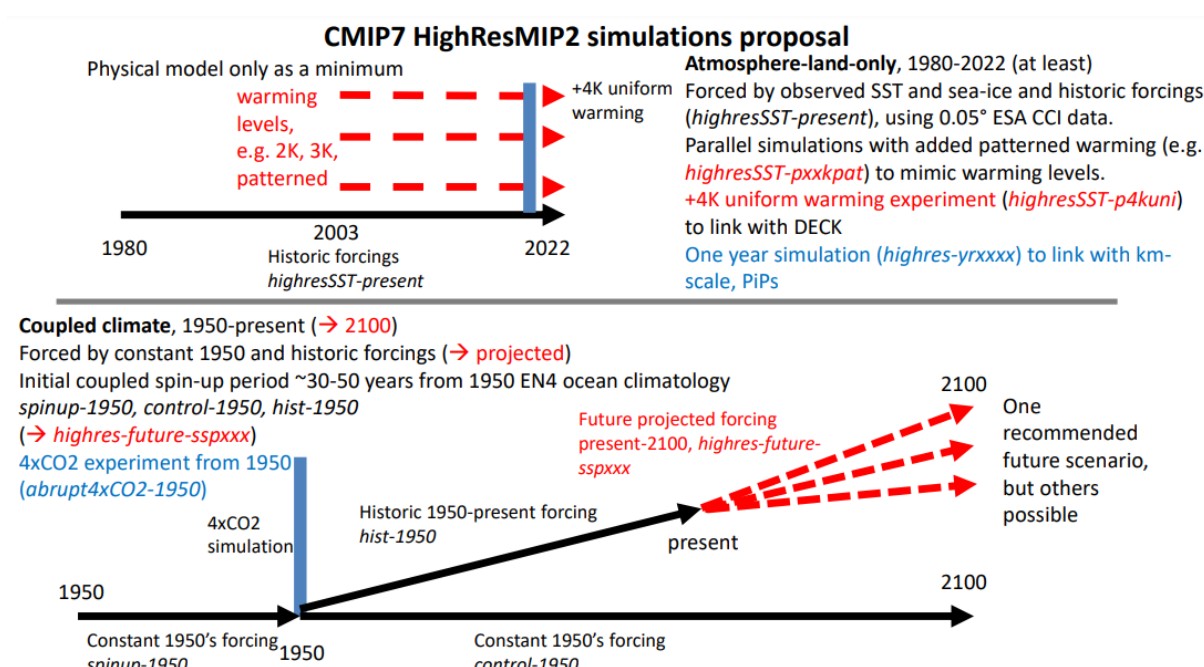

**Figure 2: Illustration of the HighResMIP2 simulations in Tiers 1-5 for (upper) atmosphere-only and (lower) coupled model experiments.**






| Tier | Experiment | Years | Total years | Desired ensemble size |
|---|---|---|---|---|
| 1a. Atmosphere-only historic | *highresSST-present* | 1980 - 2022 | 43 | 3 |
| 1b. Coupled control | *spinup-1950* | ~50 years | 50 | 1 |
| | *control-1950* | >=100 years (ideally 150 years) | 100-150 | 1 |
| 2. Coupled historical and future | *hist-1950* | 1950-2022 starting from end of *spinup-1950* | 73 | 3 |
| | *highres-future-sspxxx* xxx=CMIP7 scenario | 2023-2100+ using given future scenario, starting from end of *hist-1950* | 78 | 3 |
| 3. Idealised experiments for CMIP-comparable metrics | *highresSST-p4kuni* - uniform +4K SST AMIP-style | 1980-2022 | 43 | 1 |
| | *abrupt4xCO2-1950* - abrupt 4xCO2 starting from the end of *spinup-1950* | >= 30 years (ideally 150 years) | 30-150 | 1 |
| 4. Atmosphere-only warming levels | *highresSST-pxxkpat* pxx=warming level, e.g. +2K, +4K etc | 2003-2022 | 20 per warming level | 3 per warming level |
| 5. One year experiment | *highres-yrxxxx*, xxxx=2022 or similar historical year. *highres-yrxxxx-p4kuni,* add uniform +4K to SSTs | 1 year in historical period | 1 | 3 |

**Table 2: Proposed HighResMIP2 experiments, including Tier, name, model years, total lengths and desired minimum ensemble size.**



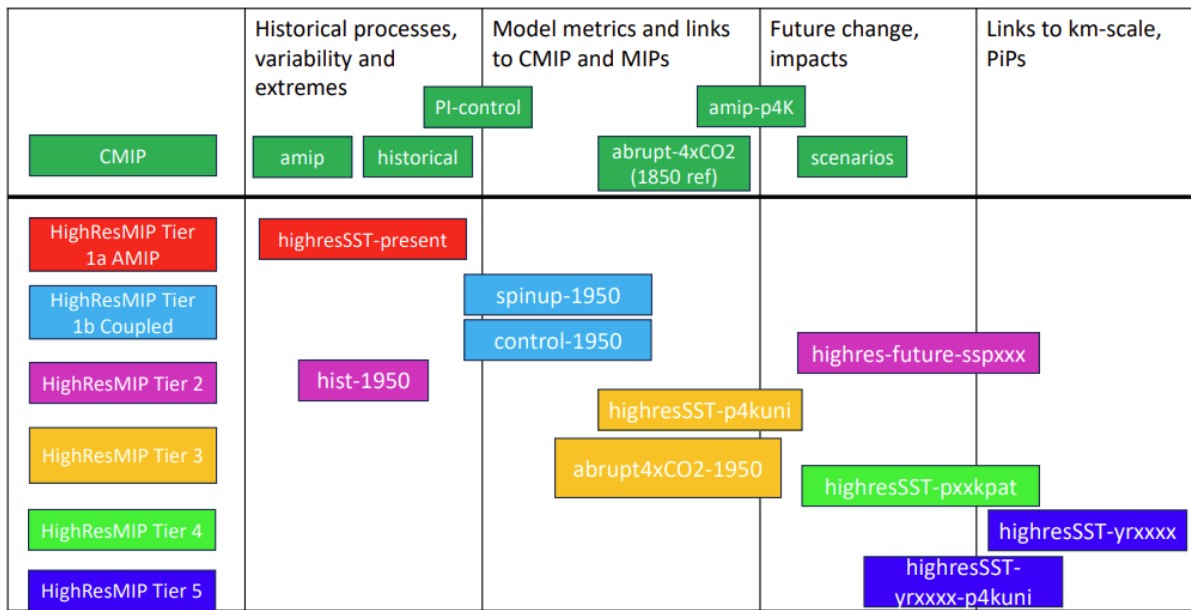

**Figure 3: Illustration of the different Tiers of HighResMIP2 simulations and how they align with applications and correspond to CMIP simulations.**

## 3.1 Detailed description of Tiered experiments

The entry card to HighResMIP2 will require at least one of the Tier 1 simulations to be completed at high-resolution, to allow groups with different capabilities (i.e., atmosphere-only or coupled modelling) to contribute, as well as some comparison to HighResMIP1. As stated before, we consider high-resolution to be 25 km or finer in both atmosphere and ocean/sea-ice components. Both Tier 1 simulations can be completed using the recommended initial conditions for atmosphere and ocean, with forcings available either via CMIP inputs4MIPs or other open access datasets (detailed below), and hence do not require any previous simulation or spin-up (Haarsma et al., 2016). Groups unable to complete the Tier 1 simulations are still encouraged to participate in other experiments, but their data will not be able to be published to ESGF as HighResMIP2.

For HighResMIP2, we do not require a parallel standard-resolution simulation (i.e. using a CMIP-class model), given the existing archive from HighResMIP1, but these will still be welcomed if produced. For coupled models, we advocate for more models with an eddy-rich ocean (~10 km resolution or finer), as there is increasing evidence that such models can provide new insights into climate variability and change (e.g., Chang et al., 2020; Zhang et al., 2022).

Providing multiple ensemble members for each simulation is very important to enable some measure of variability and uncertainty, as well as increasing the sample size of any event sets. Although one ensemble member will remain the entry



point for participating in HighResMIP2, we advocate that at least three members for each simulation would give more confidence in the results of model analysis (e.g., (Bacmeister et al., 2018; Kay et al., 2015; Roberts et al., 2020a; Rodgers et

al., 2021; Stansfield et al., 2020) and the role of external forcing, particularly given that eddy-rich oceans might enhance variability (Penduff et al., 2018). The provision of even larger ensemble sizes (likely at standard-resolution) would be welcome to enable detection of differences with a smaller ensemble of high-resolution simulations.

Given the costs involved, we anticipate that most HighResMIP2 models will not include the Earth System components to enable the simulation of a carbon cycle, and hence the following will assume the use of concentration-driven rather than

emissions-driven simulations.

*Tier 1: highresSST-present*

This is an atmosphere-only simulation similar to CMIP Atmosphere Model Intercomparison Project phase 2 (AMIP, Gates et al. 1999). The period 1980-2022 is chosen to maximise comparability with observational datasets, to align better with CMIP

AMIP, to enable new science (e.g., Meroni et al., 2023) and to enable ensembles of simulations to be produced. At the model resolutions used in HighResMIP, the resolution of the forcing has been shown to be important in the simulation of extremes (Liu et al., 2021), and hence HighResMIP retains the use of high resolution, daily forcing (in contrast to CMIP AMIP). However, we propose switching from the previous daily 1/4° HadISST.2.2.0 dataset (Kennedy et al., 2017) to the 0.05° European Space Agency (ESA) SST Climate Change Initiative (CCI) Analysis v3.0 and associated sea-ice concentration

data from the EUMETSAT Ocean and Sea Ice Satellite Application Facility (OSI-SAF) (Embury et al., 2024; Good and Embury, 2024), for both its high resolution and its regular update cycles. A comparison of the trends in surface temperature from these various products is shown in Fig. 4 (calculated following Sobel et al., 2023), as the trends have important consequences for the model simulations (Sobel et al., 2023; Zhao and Knutson, 2024) and are generally not replicated in coupled model simulations (Seager et al., 2022).






**Figure 4: Sea surface temperature trends (per time period, i.e., 36 years, 1980-2015 using annual means) for different datasets. (a) HadISST1; (b) HighResMIP dataset used in HighResMIP1; (c) ESA CCI SST dataset proposed for HighResMIP2. The common time period is constrained by HighResMIP1 only reaching 2015 with observed data, and ESA CCI only starting in 1980. The units are K / 36 years.**






We recommend using the European Centre for Medium Range Weather Forecasting (ECMWF) Reanalysis v5 (ERA5; Hersbach et al., 2020) for the atmosphere and land initial conditions for 1980-01-01. These can be obtained as described in Appendix A. The first two years 1980-81 will be considered as spin-up for the land surface, so detailed analysis should be done on the period 1982-2022. Other forcings are the same as for CMIP7 AMIP. However, we recognise that two years may

be too short for the land surface to spin-up, and so alternative methods to produce its initialisation are acceptable but should be fully documented.

*Tier 1: spinup-1950*

This experiment provides a multi-decadal spin-up for the coupled model simulations and its final output is used as initial conditions for both *control-1950*, *hist-1950*, and *highres-4xCO2*. It is essentially the same as in HighResMIP1. Most groups

in HighResMIP1 ran the *spinup-1950* simulation for 50 years starting from initial conditions defined in Appendix B. There are trade-offs between the length of this simulation (to produce a more equilibrated ocean), against the non-zero Top-of-atmosphere (TOA) in 1950 providing net heating of the ocean and hence forcing it away from its initial conditions. Other methods to provide improved ocean initial conditions and spin-up are being sought, led by the WCRP ESMO (Earth System Modelling and Observations) group (World Climate Research Programme ESMO, 2024).

All forcings should be set to values representative of a constant 1950's state, but the precise form may vary by forcing. For solar, ozone, it is suggested to produce a 11-year climatology centred on 1950 to average the solar cycle. Greenhouse gas (GHG) concentrations can simply use the values reached in 1950 as defined in CMIP. For aerosol emissions and volcanic forcing, a mean over 10 years is suggested around 1950 (i.e., 1946-1955 if available, or 1950-1959 if not).

*Tier 1: control-1950*

This will be a backward-compatible simulation design (though with updated forcings) to HighResMIP1 and remains a valuable way to gauge model variability without changes in interannual forcing. It is initialised from the end of *spinup-1950*, with constant 1950's forcings as used there.  It will be essential to compare the optimised high-resolution model versions in HighResMIP2 to those submitted previously in HighResMIP1. This simulation should be run for at least 100 years, ideally 150 years if the future scenarios are also going to be produced (see below), to act as a reference to the simulations with

varying forcing and hence enable some comparison of model drift against forced changes.

*Tier 2: hist-1950*

This is similarly backward-compatible to HighResMIP1, and is initialised from the end of *spinup-1950*. It uses time-varying forcings from CMIP7, mimicking the CMIP7 historical simulation but running from 1950 to 2022.

*Tier 2: highres-future-sspxxx*

This is the coupled future projection simulation, and is initialised from the end of the *hist-1950* simulation. HighResMIP1 only used the "high emissions" future scenario SSP5-8.5 in the simulation design, but given recent policy pledges and actions, the current level of CO2 emissions is well below this scenario (Meinshausen et al., 2024), making it less attractive for use. Factors influencing our choice include: the need to explore plausible worst-case scenarios; examining the




implications of a wide-range of warming levels, including the possibility of tipping points; a desire for enhanced
collaboration with other groups (e.g., CORDEX used SSP3-7.0 as one of their scenarios) and advice from the CMIP Panel.
Noting that future scenarios for CMIP7 are still in development, we propose to recommend a CMIP7 future scenario that will
produce an additional radiative forcing, at the end-of-century, of around 7 W m$^{-2}$. Simulations using other scenarios will still
be accepted, but a larger multi-model ensemble using this preferred scenario would enable more coordinated science.

We propose adding several idealised simulations to mirror those in CMIP DECK, so that we can build a comparison of
metrics between DECK and HighResMIP2 and hence have some characterisation of HighResMIP2 models.

*Tier 3: highresSST-p4kuni*

To this end we propose a uniform +4K experiment. This simulation is parallel to *highresSST-present*, using all the same
forcings apart from SST, with +4K added to all SST points uniformly. It is analogous to the CFMIP experiment *amip-p4k*,
but using the HighResMIP2 SST forcing rather than AMIP. It will be used to look at climate feedbacks and precipitation
responses, with further work needed to build a correspondence between metrics from this simulation and comparable ones
from CMIP/CFMIP simulations.

*Tier 3: abrupt4xCO2-1950*

In addition, we propose an abrupt 4xCO2 simulation. This coupled simulation starts from the end of the *spinup-1950*
experiment, and enforces an instantaneous increase of 4xCO2 (analogous to the CMIP experiment *abrupt-4xCO2*) but
referenced to 1950, and parallel to the *control-1950* simulation. Evidence suggests (Dong et al., 2020) that for the abrupt
4xCO2 simulation, years 1-20 of simulation produces metrics (such as Effective Climate Sensitivity, EffCS) that strongly
correlate with the longer term (21-150 year) feedback, but may be considerably smaller in magnitude. Given the expense of
high-resolution simulations, just 20-30 years of this simulation could produce useful information for the community. Further
work will be needed to understand the relationship between the 4xCO2 experiment referenced to 1850 (as in CMIP) and
1950 (as here in HighResMIP), and the role of ongoing model drift. An illustrative example comparing the EffCS calculated
from CMIP and HighResMIP simulations using the same model science configuration is shown in Fig. 5.



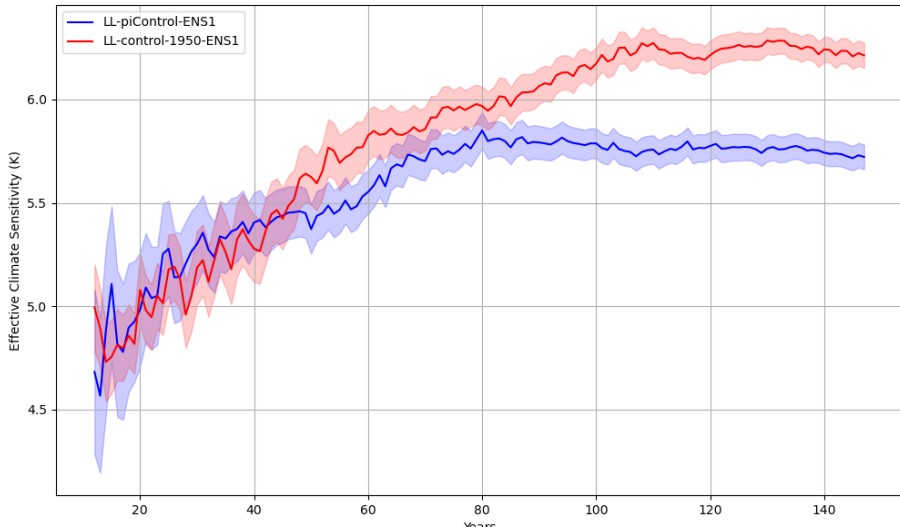

**Figure 5: An illustrative example of model metrics enabled by new HighResMIP2 simulations: Effective Climate Sensitivity**
**490 (EffCS) calculated with successively more years of data for two simulations, in this case a ~130-km atmosphere coupled to a ~100-km ocean. Blue uses the standard CMIP6 DECK piControl and abrupt-4xCO2 simulations to calculate EffCS over 150 years; Red uses the standard HighResMIP experimental design, 50 years of spinup-1950, and then control-1950 and abrupt4xCO2-1950 simulations. Using only 30 years of simulation clearly does not define the final (150 year) EffCS, but may at least give a useful lower bound. Any relationship across experimental designs will need more (multi-model) analysis.**


*Tier 4: highresSST-pxxkpat*

In order to make the SST-forced simulations more clearly actionable than in HighResMIP1, we propose to create different global warming level experiments to be run in parallel with the *highresSST-present* simulation. Such atmosphere-only experiments use patterned SST (and sea-ice) changes imposed on the observed 2003-2022 period (a 20-year subset to reduce
the global trend within the period), with the globally-averaged warming levels of surface air temperature calculated above the historical values, and with appropriate changes to other forcings. The approach has been demonstrated by the "Half a degree Additional warming, Projections, Prognosis and Impacts" (HAPPI) experiment (Mitchell et al., 2017; Wehner et al., 2018) and the database for policy decision-making for future climate changes framework (Ishii and Mori, 2020). Similar to the Global Warming Levels methodology used in IPCC AR6 for global surface air temperature, these simulations can be
linked to global mitigation policy targets and are shorter and less computationally expensive than fully coupled integrations, thus permitting ensembles of simulations. Structural uncertainty in SST changes (and consequences for climate impacts) can be directly examined by utilising multiple perturbed SST datasets constructed from individual large ensemble CMIP-class models or the entire CMIP database, or by explicitly encompassing different SST trend patterns such as in the tropical Pacific. Such simulations enable the study of impacts for a given warming level in a simplified atmosphere-only framework,



hence providing improved links with CMIP and insight into climate extremes processes. It is worth noting that the use of warming level experiments is commonplace in event storyline simulations as well (e.g., Huprikar et al., 2023).

*Tier 5: highres-yrxxxx* (and *highres-yrxxxx-p4kuni*)

These simulations are expected to be one year in duration, short enough to be feasible for km-scale models (finer resolution

than Tier 1) as well as more typical climate resolutions (including those used for Tier 1 and CMIP7 more broadly), and hence enabling comparisons across classes of models. This will be designed to be as consistent with the DYAMOND3 protocol as possible (Takasuka et al., 2024), and is envisioned to enable faster testing and experimentation (compared to the Tier 1-4 simulations), and to allow assessment of basic aspects of climatology and daily-sub-daily aspects of variability. It is likely to be atmosphere-only with prescribed SSTs and sea ice. Groups could consider using this simulation to test further

increases in their underlying model component resolutions that approach those of DYAMOND. The preferred year is March 1, 2020-February 28, 2021 to match that used in DYAMOND3 and it lies within and towards the end of the time range of *highresSST-present* in Tier 1, in order to make most use of the newest observational datasets. Additional years could be added (e.g., to accompany new satellites such as EarthCARE, Illingworth et al., 2015). A uniform +4K experiment (*highres-yrxxxx-p4kuni*) following Cess and Potter (1988) is also likely, enabling a chain of metrics on climate feedbacks to be

evaluated sequentially from this one year simulation, to the longer *highresSST-present*, to AMIP and hence CMIP. Pairing of the historical and +4K simulations would allow us to address questions related to the sensitivity of cloud feedbacks to model resolution, and related topics.

## 4 Data requirements

As discussed in Haarsma et al., (2016), the output, storage, and publication of high-resolution model data are challenging issues. As the resolution of HighResMIP1 models approaches the scales necessary for realistic simulation of synoptic and mesoscale phenomena, daily and sub-daily multi-level data (in both ocean and atmosphere) are of increasing interest to allow the investigation of weather phenomena such as those related to mid-latitude storms, blocking, tropical cyclones, and monsoon systems as well as ocean mesoscale processes and ocean extremes such as marine heatwaves. In addition, they

could be valuable data to train or evaluate new machine-learning (ML) models and approaches. However, increased data outputs can have significant impacts on: model speed due to I/O operations; the size of data archives to enable storage; the challenge of formatting via the Climate Model Output Rewriter (CMORisation of variables); and on publication to ESGF due to data volumes. Feedback from modelling groups after HighResMIP1 suggested the data volumes requested were narrowly manageable and data providers would not want to have them grow significantly. In the following we discuss

options for managing data volumes.



The Data Request (DReq), the variables and frequencies that are required and suggested to be produced from a CMIP MIP to answer its science questions, is being developed for HighResMIP2, and will be informed by CMIP7 plans following on from CMIP6 (Juckes et al., 2020). The EU PRIMAVERA project undertook a study of the variables downloaded from the HighResMIP1 archive (either from ESGF or within the PRIMAVERA project over a six month period), and produced tables

of data access by volume and by frequency (this is available at Seddon et al., 2020). Together with community engagement, we have used these tables as a basis for an initial updated Data Request for HighResMIP2, including prioritising the variable and frequency based on their access. We kept all monthly data requests the same as for HighResMIP1, and have listed the top 60 downloaded variables as priority 1 data as well. In addition, as part of the EU EERIE project (EERIE, 2024), we have added specific extra tables to the Data Request for high frequency outputs (Savage, 2023), but the applicability of this for the

broader HighResMIP community will need to be reviewed. It is possible that limiting output variables in this way may restrict future novel analysis ideas, but the additional experiments suggested for HighResMIP2, specifically the shorter simulations, may give more scope for additional outputs tailored to these simulations. Based on the recommendations of the metrics section (Section 5), we will prioritise variables that can contribute to key metrics and assessment packages to be used for evaluating HighResMIP2 simulations (and CMIP models more generally).

By implementing the DReq in this way, we aim to keep the number of variables at a manageable level. However, the data volumes are also determined by the spatial resolution, precision and compression at which the variables are stored. Although the DReq states where native grid data is required, making such choices depends on the properties of the data and underlying processes, as well as what information is relevant for end users and the available observational products for comparison and evaluation. Generally, the most relevant data for impacts is at the planetary surface where observational datasets have the

highest resolution. It is therefore desirable to store surface and near-surface variables at higher temporal and spatial resolutions than elsewhere. Furthermore, in order to evaluate the HighResMIP ensemble, the high-frequency output should contain variables for which high-frequency observations are available as well.

Guidance by the CMIP6 Panel and WGCM Infrastructure Panel on model grids (e.g. Griffies et al., 2016; CMIP6 Output Grid Guidance, 2024), will be used to help inform our choices. Consideration of effective model resolution (e.g. Klaver et

al., 2020), based on kinetic energy spectra, can also help to inform choices about the "optimal" resolution to share data generated by HighResMIP2, and we encourage modelling groups to calculate and publish this metric to inform data users.

Models using unstructured meshes are becoming increasingly common, which produces new challenges for both data providers and users. Using conventional CMOR data standards can greatly increase the published volumes of such data (due to grid descriptors in every file), and so HighResMIP2 will (informally) experiment with ways to address this issue in

addition to standard CMIP formats. There is a growing ecosystem to handle unstructured grids (e.g. UXarray Organisation, Chmielowiec et al., 2024), and we encourage data users to consider building new workflows to take advantage of these tools rather than simply regridding data to regular latitude-longitude meshes to fit with existing analysis codes. Using native grid data may also obviate producing a duplicate of the data on an alternative grid, hence saving storage and processing. For some analyses, there is no substitute for using data on the native model grid, particularly when calculating integrated transport



quantities where conservation is important or for some extreme events. Preserving as much data on native grids as feasible also maintains provenance and improves transparency and reproducibility by mitigating data loss associated with post-processing.

However, regridding to standard or target grids can still play an important role in some analyses, particularly when confronted with decisions in the face of limited resources. Away from the planetary surface, it may be more valuable to

provide high-frequency output at a "useful" resolution (e.g. lower spatial resolution, but hourly) rather than making no compromises on horizontal grid spacing but having to limit temporal resolution (e.g., higher spatial resolution, but daily). Using standard grids can also be sufficient to produce standard metrics (see Section 5), fit into existing workflows and be more comparable to existing observational datasets on regular grids.

If data volumes really become so large that they are impractical to work with (e.g. at the timestep level), then performing

such calculations online (i.e. either within the model itself or as part of post-processing) may be much preferred for accuracy. A simple example is integrated vapour transport (IVT) which is commonly used to track atmospheric rivers (Section 5). IVT, at any given time, is a two-dimensional field that is the vertical integral of multiple three-dimensional fields. Calculating such variables as far "upstream" as possible reduces the data burden on scientists and other end users of HighResMIP2 data.

Data compression is another promising avenue to keep data volumes manageable regardless of output spatial and temporal

resolution. This can range from relatively simple improved lossless compression settings in CMORised netCDF files that provide marginal filesize reduction, to lossy compression methods that reduce data storage more aggressively at the expense of bit-for-bit reproducibility (Baker et al., 2016; Klöwer et al., 2021). We will be advised by CMIP7 on any plans for enhanced data compression, as any method would still need to conform to CMOR standards.

An alternative way to reduce the data volumes is to only output a more demanding set of data over sub-periods of the full

simulation (e.g., time-slice method). In an eddy-rich ocean model, daily or 3-daily three-dimensional output is needed to analyse the variability of western boundary currents and sub-surface eddies and to estimate the eddy contribution to the meridional overturning circulation in density space in z-coordinate models. To understand processes leading to TC development, sub-daily three-dimensional output and a variety of sub-daily radiative and surface flux data is needed to quantify cloud-radiative and surface flux feedbacks (Dirkes et al., 2023; Wing et al., 2019). Such time-slice methods can

enable more detailed process-based analysis while not overwhelming processing capacity. A downside of this method is that it is more complicated to set up and run the simulations. We propose that for coupled simulations there are enhanced data output periods near the start, around present-day, and towards the end of the simulation (e.g., 1950-1960, 2010-2020, and 2090-2100), where we encourage modelling groups to produce more variables outside of priority 1 (which we expect to be produced throughout). The different Tiers of simulation in HighResMIP2 may also require an amended DReq, for example

in the Tier 5 one year simulations where the focus will be on higher frequency processes.

We also encourage the modelling groups to produce more derived diagnostics for HighResMIP2, which to some extent can obviate the need to publish all the high frequency outputs at native resolution. Various analyses in HighResMIP1 made use of derived datasets that were created from multi-model output and then separately made available to the community (e.g.,



model TC tracks (Roberts, 2019) and synthetic tracks (Bloemendaal et al., 2022); sea level and storm surge (Muis et al.,
2023); wind storm footprints (Lockwood et al., 2022)). As discussed in Section 5, there are several sets of algorithms and
parameter settings that, if used by the modelling groups, could lead to extremely valuable datasets that have a common
baseline and could be published. For such datasets to be published to ESGF might require some refinement of data standards,
but this should be minimal as long as the variables within the derived dataset follow CMOR standards.

One strong recommendation from our past experience is to take the analysis tools to the data, where possible, in preference
to moving the data. An example of this would be the UK CEDA-JASMIN platform (Seddon et al., 2023), where many of the
simulations from HighResMIP1 were published and where collaborators could obtain access to the systems and analyse the
data. Similarly, a large chunk of the HighResMIP1 archive was copied once to the U.S. National Energy Research Scientific
Computing Center (NERSC, W. Collins, personal communication), enabling many US collaborators to access and analyse
the data without further duplication. This was facilitated via the availability of Globus at JASMIN, enabling access to high-
speed data transfer nodes and providing transfer speeds 100x faster than available via ESGF.

## 5 Towards standard metrics and diagnostics

In order to make use of HighResMIP2 simulations to advance the scientific questions outlined, we propose a standard set of
metrics and diagnostics of both the mean climate and individual processes of interest.  The various metrics and diagnostics
that are anticipated to be used are summarised in Table 3. We also build on experiences and lessons-learned from the
analysis and advances from HighResMIP1.

| Phenomena | Preferred Software Package |
|---|---|
| Mean Climate | PMP |
| Precipitation | PMP |
| ENSO | PMP |
| MJO | PMP |
| Extratropical Variability | PMP |
| Monsoon | PMP |
| Tropical Cyclone | TempestExtremes |
| Extratropical Cyclones | TempestExtremes |
| Atmospheric Rivers | TempestExtremes |
| Mesoscale Convective Systems | TempestExtremes |
| Ocean Eddies | py-eddy-tracker |

**Table 3: Summary of planned metrics and diagnostics for HighResMIP2.**




Routine benchmarking used for ESMs' metrics are needed for each modelling group to quantify potential reductions in existing biases, as well as for addressing stakeholder needs and applications (Reed et al., 2022). To this end, and following the CMIP7 Model Benchmarking Task Team, HighResMIP2 can make use of the PCMDI Metrics Package (PMP) (Lee et al., 2024). PMP is a publicly-available Python software package that has been developed to provide objective comparisons of

climate and Earth system models with one another. Thus HighResMIP2 participants can readily compare their high-resolution simulations with conventional versions of their modelling systems that have been submitted to CMIP6 and CMIP7, including DECK simulations. PMP includes typical metrics for mean climate, at the global and regional scales, that allow for comparisons between simulations, as well as between a simulation and observations (or reanalysis), for a variety of variables, including precipitation, sea level pressure, radiative fluxes, winds, and temperature. These metrics use traditional

measures like bias, root-mean-square differences, and pattern correlations. Additional metrics for precipitation, including extremes, drought, and seasonal and daily cycles are also included. The PMP also contains standard metrics for important modes of climate variability, including El Niño-Southern Oscillation (ENSO), Madden-Julian Oscillation (MJO), and monsoon characteristics in various regions. In addition, PMP quantifies various important modes of extratropical variability, including the Northern Annular Mode (NAM), the North Atlantic Oscillation (NAO), the Southern Annular Mode (SAM),

the Pacific North American pattern (PNA), the North Pacific Oscillation (NPO), the Pacific Decadal Oscillation (PDO), and the North Pacific Gyre Oscillation (NPGO). Finally, the package also includes a standard metric for cloud radiative feedbacks, including using *abrupt4xCO2-1950* Tier 3 experiment for estimating climate sensitivity.

In addition to documenting the output of PMP, we ask that participating modelling groups also provide monthly output of standard mean climate fields on a 0.25 deg. latitude-longitude grid for each simulation (though details are left to the typical

approach by modelling groups for archiving CMIP output). This output could then be used to apply other variability and mean climate metrics as needed, but also significantly cuts down on the data submission requirements. Given that this is monthly data on a coarsened grid, we anticipate that these will be provided for all simulations.

Process-level analysis will be a critical component of HighResMIP2 science activities. Following HighResMIP1, individual events such as TCs, extratropical cyclones, atmospheric rivers, and MCSs will be studied. Such events are often under-

resolved or not captured at all in traditional CMIP-class models as demonstrated in HighResMIP1 activities. Given the large-data volumes typically required for objective tracking at high temporal resolutions, as well as the advantages of tracking such events on the native grids of models, HighResMIP2 can make use of the open-source TempestExtreme package (Ullrich et al., 2021). To identify and track these features output will be required at frequent intervals of at least 6 hourly using the specific parameters and thresholds documented in Ullrich et al., (2021) for TCs, extratropical cyclones, and atmospheric

rivers and in Hsu et al., (2023) for MCSs, as is common for model evaluation (e.g., Reed et al., 2023). Ocean features will also be a focus of HighResMIP2. It is expected that ocean eddies, and their role in the climate system, will become more prominent in HighResMIP2 simulations as model resolution increases (see Table 1). To detect and track eddies, the py-eddy-tracker toolbox will be used (Mason et al., 2014). Additional ocean extremes such as marine heatwaves (MHW) would also be welcome to be provided as outputs following an agreed method such as Hobday et al., (2016).



While it is strongly encouraged that modelling groups provide the feature characteristics and trajectories from these specific software packages, modelling groups that are interested in submitting trajectories from other techniques are welcome to do so to complement the standardised forms. In all cases, we recommend that the features be tracked for the full length of all simulations, but understand that in some cases modelling groups may need to prioritise feature tracking for a subset of the simulations given data loads.

In addition to the above metrics and diagnostics, as part of the Data Request for HighResMIP2 it is asked that the modelling groups also provide a standard set of 2D output for each simulation that could be used to help inform studies related to future adaptation and climate risk planning. In particular, precipitation, near-surface temperature, surface pressure, and surface winds at 6-hourly temporal resolution for the historical and future (both AMIP and coupled) are requested on a 0.25 deg. latitude-longitude grid, as discussed earlier, or the native grid of the model. This output could be used for additional

extremes analysis, as well as to force other downscaling or relevant hazard models (e.g., storm surge, flooding, etc.), that focus more on climate impacts. By comparing different future scenarios and warming levels scientists and practitioners might be able to better inform resilience measures at regional-to-local scales given the high resolution of HighResMIP2. Again, it is expected that these 2D fields be provided for all simulations, but understand that some model groups may need to prioritise a subset of the simulations, or time periods.


## 6 Summary and Discussion

CMIP6 HighResMIP was the first global high-resolution model intercomparison as part of a CMIP effort, and as such it continues to expand our understanding of the role of grid-resolution in multi-model climate simulations, including new insights into large-scale variability and trends as well as extremes. With the rapid developments in this arena (e.g., via

improved and optimised models capable of being deployed at global storm-permitting and global km-scales) over the last decade, as well as the fast evolution of machine-learning methods and requirement for training data, there is a demand for a new set of global high-resolution simulations. We do recognise the considerable cost and carbon footprint of such high-resolution simulation, both in terms of supercomputing and storage, but consider that the new information and capabilities that they enable justify the expense.

The key advancement for HighResMIP2 will be to use models optimised for high-resolution, rather than being constrained to be as similar as possible to standard-resolution models. Together with anticipated increases in model resolution to better represent processes and address climate projection uncertainties (specifically towards ocean eddy-rich resolutions), and relaxation of other constraints from HighResMIP1 (such as idealised aerosols), we believe that our new HighResMIP2 protocol will further advance our understanding of climate. Together with updated and new simulations, data requirements

and metrics, HighResMIP2 outputs will be capable of making important contributions to impact and hazard modellers, climate risk assessments, and policy.



Our new idealised simulations should also enable improved characterisation of HighResMIP models and hence enable improved links to CMIP-class models, as well as giving some extra flexibility, such as encouraging the use of older models (already tested at high resolutions) rather than having to await new CMIP7 model versions.

The development of the approach for HighResMIP2 documented has been a collaborative process among the authors, the HighResMIP2 working group members, the CMIP panel, and the broader community. As a result, it is expected that there will be a natural integration of HighResMIP2 analysis and existing efforts within CMIP and the WCRP. An obvious example will be the comparison of the HighResMIP2 simulations to the CMIP7 DECK simulations, which will directly advance the stated science questions in Section 2. Another example is related to CORDEX as a leader in utilising regional models to

downscale CMIP output to inform regional climate applications. This provides an opportunity to coordinate analysis approaches between CORDEX and HighResMIP2, allowing for the investigation of the relative strengths of the different approaches to regional climate and extreme event simulation. Furthermore, willing modelling centres could consider using HighResMIP2 output to drive their CORDEX models, which could further advance the scientific questions and shed light on potential approaches to high-resolution in future intercomparisons. In the broader context, HighResMIP2 will complement

other CMIP activities and help advance the objectives of many of the WCRP lighthouse activities.

Data volumes are an ongoing concern for HighResMIP2 (and the community more generally), and we will continue to engage with others, including the Fresh Eyes on CMIP group, to understand which model variables at what frequencies are the most valuable to produce, and with other groups on new methods for reducing data sizes.

However, although model data at high spatial and temporal resolutions are difficult to manage and store, we must recognise

that such data is important. Understanding the weather of the future (not just the climate) will be a key to adaptation planning and risk reduction. It will also become increasingly important with the rapid advance of machine learning technologies. The development of sub-grid parameterization processes, partly and fully utilising these technologies, has significantly improved the accuracy of climate and Earth system models while also reducing computing costs. Correction techniques by machine learning and deep learning can use historical HighResMIP dataset outputs as training data to improve future climate change

projections, similar to the correction method used in numerical weather forecasting models. HighResMIP data could also be used to develop and challenge existing and new ML-based weather and climate models, ranging from testing specific processes through to use as global training data, and hence help the community to better understand what the weather of the future will look like.

**Appendix A: Initial conditions for *highresMIP-present***

The atmosphere (and land and lakes if required) initial conditions for 1980-01-01 come from ERA5 reanalysis (Hersbach et al., 2020) on that day. These can be obtained via the Copernicus Climate Change Service (C3S) Climate Data Store (CDS)





(DOI:10.24381/cds.adbb2d47) by constructing an API script using the https://cds.climate.copernicus.eu/api-how-to and variable selection from

https://apps.ecmwf.int/data-catalogues/era5/?stream=oper&levtype=ml&expver=1&month=jan&year=1980&type=an&class=ea

The SST and sea-ice forcing for *highresSST-present* comes from the ESA CCI dataset at version 3.0, and can be obtained from Good and Embury (2024). This dataset does not include temperatures for inland lakes, so for models without their own

lake parameterisations, lake temperatures can for example be obtained from the OSTIA dataset https://data.marine.copernicus.eu/product/SST_GLO_SST_L4_REP_OBSERVATIONS_010_011/description

For the initialisation of ensembles, we recommend these be generated via perturbations of initial conditions rather than perturbation of forcing fields.

**Appendix B: Initial conditions for *spinup-1950***

The initial conditions for the coupled *spinup-1950* experiment, by default, remain the same as in HighResMIP (Haarsma et al., 2016): ERA-20C reanalysis for atmosphere and land (Poli et al., 2016), and EN4 ocean analysis for ocean temperature and salinity (Good et al., 2013), specifically version EN.4.2.2.g10, taken as a climatology around 1950. The sea-ice can be initialised in any appropriate way. While we acknowledge that different ocean analysis products and other developments

(e.g. Hermanson et al., 2023; Karspeck et al., 2017) may give "better" initial conditions, using a common approach will make analysis and comparison of the multi-model dataset much more straightforward.

**Data and code availability**

The SST data used to construct Fig. 4 is freely available: HadISST1 (Rayner et al., 2003) from

https://www.metoffice.gov.uk/hadobs/hadisst/data/download.html; HighResMIP1 data, from inputs4MIPs (Kennedy et al., 2017); ESA-CCI SST data, from Good and Embury (2024). The trend files and code that produced the figure are available from doi:10.5281/zenodo.13312568, and follows Sobel et al., (2023).

The data and code used to produce Fig. 5 is available also from the doi:10.5281/zenodo.13312568. These include timeseries of annual mean surface air temperature (TAS) and Top of Atmosphere radiation (TOA) from the three model simulations.


**Author Contribution**



The HighResMIP co-leads MJR, KAR and QB conceptualized the structure of the manuscript and led the writing of the original draft, as well as developing the visualizations of HighResMIP2. All other co-authors contributed to review and editing of the manuscript. MJR, ML provided software for Fig. 4, 5.


**Competing interests**

The authors declare that they have no conflict of interest.

**Acknowledgements**

Support for this work was provided by numerous funding sources and are summarised here (grouped by author's initials
and/or funder):

MJR, HMC, ML, JS, PLV were supported by UK Research and Innovation (UKRI) under the UK government's Horizon Europe funding guarantee (grant numbers 10057890, 10049639, 10040510) as part of EU EERIE. MJR, HTH, ML were supported by the Met Office Hadley Centre Climate Programme funded by DSIT. KAR was supported by grants (NSF AGS2217620, NSF AGS2244917, Dept. of Energy DE-SC0016605, and Dept. of Energy DE-SC0023333). QB was funded
by the National Key Research and Development Program of China (grant no. 2022YFF0802003). LPC and DP acknowledge the support of the Ministère des Relations internationales et de la Francophonie. SJC was supported by the National Science Foundation (AGS-2043142, AGS-2244918), the NOAA MAPP program (NA21OAR4310345, NA23OAR4310600, NA23OAR4310598) and the NOAA CVP program (NA22OAR4310610). PC and GD were supported by the National Science Foundation grant number AGS-2231237 and the National Academies of Science and Engineering Gulf Research
Program grant number 2000013283. HMC was funded by Natural Environment Research Council grant number NE/P018238/1 and through a Leverhulme Trust Research Leadership Award. For GD, the US National Science Foundation (NSF) National Center for Atmospheric Research (NCAR) is a major facility sponsored by the US NSF under Cooperative Agreement 1852977. IF was funded/co-funded by the European Union (ERC, OSTIA, 1 101116545) - views and opinions expressed are however those of the author(s) only and do not necessarily reflect those of the European Union or the
European Research Council. Neither the European Union nor the granting authority can be held responsible for them. NSF was funded by the European Union's Horizon 2020 research and innovation programme under the Marie Sklodowska-Curie grant agreement No. 846824, National Institute for Health and Care Research award NIHR204850, and Natural Environment Research Council grant number NE/Y503319/1. SH acknowledges support from the Deutsche Forschungsgemeinschaft (DFG, German Research Foundation) under Germany's Excellence Strategy-EXC 2037 "CLICCS-Climate, Climatic
Change, and Society"-Project Number: 390683824, contribution to the Center for Earth System Research and Sustainability (CEN) of Universität Hamburg. DK was funded New Faculty Startup Fund and Creative-Pioneering Researchers Program from Seoul National University, the National Research Foundation of Korea (NRF) grant funded by the Korea government (MSIT) (RS-2024-00336160), NASA MAP program (80NSSC21K1495), NOAA MAPP program (NA21OAR4310343),



and NOAA CVP program (NA22OAR4310608). CK was supported by JSPS KAKENHI Grant Number JP20H05728 and by
MEXT as "Feasibility Studies on Next-Generation Supercomputing Infrastructures." LRL was supported by Office of
Science, U.S. Department of Energy Biological and Environmental Research as part of the Regional and Global Model
Analysis program area; PNNL is operated by Battelle Memorial Institute for the U.S. Department of Energy under Contract
DEAC05-76RL01830. RM was supported by MEXT-Program for the advanced studies of climate change projection
(SENTAN) Grant Number JPMXD0722680734. PN was funded by INCT-MC2 under Grants: CNPq 465501/2014-1,
FAPESP 2014/50848-9, and CAPES 88887.136402/2017-00, and 88887.115872/2015. PO was supported by the Spanish
fellowship RYC-2017-22772. CDR and AMT were supported by the EERIE project (Grant Agreement No 101081383)
funded by the European Union as part of Horizon Europe. ES was funded by EU project BlueAdapt under Grant agreement
101057764. CYT was funded by NSTC: 112-2111-M-001-010 and Academia Sinica: AS-GCP-112-M03. PAU was
supported by the Program for Climate Model Diagnosis and Intercomparison (PCMDI) under the auspices of the US DOE by
Lawrence Livermore National Laboratory under Contract DE-AC52-07NA27344. PLV was supported by the NCAS core
grant. MFW was supported by the Director, Office of Science, Office of Biological and Environmental Research of the U.S.
Department of Energy under Contract No. DE340AC02-05CH11231 under the Regional and Global Model Analysis
(RGMA) program. CZ acknowledges support of the US Dept. of Energy (DE-SC0016605 and "Integrated Coastal
Modeling"). BZ is supported by the NOAA Climate Program Office through Climate Process Team 140Y8R1ES2/24P01;
the statements, findings, conclusions, and recommendations are those of the author(s) and do not necessarily reflect the
views of the National Oceanic and Atmospheric Administration, or the U.S. Department of Commerce. WZ acknowledges
the support from NOAA (Award NA23OAR4310611- T1-01), Utah Department of Natural Resources (Award 205396) and
the U.S. Geological Survey under Grant G24AP00051-00. We also thank Timothy Andrews (Met Office) for the original
code for Fig. 5.

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
