# Peer review of "High Resolution Model Intercomparison Project phase 2 (HighResMIP2) towards CMIP7"

_EGUsphere, 2024_

## Author Response (AR1)

HighResMIP2 reviewer comments (revised)

**Reviewer 1**

Review of "High Resolution Model Intercomparison Project phase 2 (HighResMIP2) towards CMIP7" by Roberts et al.

The paper presents a discussion of the second phase of the HighResMIP initiative (HighResMIP2). The paper motivates the need for a second phase, describes the requested experiments, and lays out the current thinking regarding the model output and data storage. Overall, there is nothing fundamentally wrong with the paper, and a dedicated paper that describes HighResMIP2 is of course needed. As such, the paper will make an important contribution to GMD.

That being said, I find the paper could be substantially shortened and streamlined, and this could make the paper more accessible and increase its impact. Several issues are repeated multiple times, for example the issue of ocean eddies, and I am not sure what sentences like "We do recognise the considerable cost and carbon footprint of such high-resolution simulation, both in terms of supercomputing and storage, but consider that the new information and capabilities that they enable justify the expense.", which is in the conclusion section, add to the paper. Other examples of in my view unnecessary text are L636-639, and much of the text in section 4.

We thank the reviewer for their time and the very useful comments and suggestions.

Response: As Reviewer 3 points out there are benefits of extensive dialogue and inclusive collaboration of scientists. However, one limitation is that descriptions can become a little verbose and repetitive.  We will work through the manuscript to reduce repetition and use more concise phrasing when possible.
We have reduced some of the repetition (e.g L232, L261, L315).

However, because this is a protocol paper that is meant to be a reference for the modeling group completing simulations and individual scientists and students analyzing the output of the simulations, we believe that it is important to provide detail.  Also, we believe that some repetition across the different sections is needed if a reader is focusing on one aspect.  A good example of this is Section 4, where we believe the level of detail will be useful for both modeling groups and analysis teams.

Specific comments:

L114: In what sense is the prediction skill improved? In terms of reproducing the historical climate, in terms of hindcast decadal predictions, or something else?

Response: We have modified this to "with implications for decadal variability and climate projections." HighResMIP1 simulations themselves were not part of, e.g., CMIP6 decadal predictions, so we cannot talk about prediction skill directly. It is true that the model resolutions referenced (and used for decadal hindcast predictions) are consistent with those used in HighResMIP1, and show improved skill, but we leave this out.

L146-148: I found this sentence harder to read than the rest of the text.

Response: We have updated the text to: "It was therefore important to give modelling groups (see Table 1) an early sign of the HighResMIP2 simulation design (prior to CMIP7 forcings being available), so that they could make informed choices about when to start new simulations, and how HighResMIP1 and HighResMIP2 simulations would relate to each other."

around L235: Are there any recommendations for modeling groups regarding the treatment of deep convection? For example, ICON in the nextGEMS setup disables the deep convection scheme entirely, whereas IFS-FESOM still uses it, though in a much reduced form.

Response: We leave the configuration of and model design choices for the models, including the treatment of deep convection, to the modelling groups to make their own decisions. This is for two main reasons (1) this is consistent with the approach of the broader CMIP community and (2) we don't think there is a community-wide consensus on such choices for deep convection. We have added a sentence at the end of this paragraph: "We emphasise that choices of model physics and dynamics with resolution (such as treatment of deep convection) are left with the modelling groups." Part of the reason for the one year experiment is an arena to test and document such changes relatively cheaply.

L252ff: The arguments put forward here for prescribed-SST simulations equally apply to coupled simulations in my view.

Response: This is a good point.  At the end of this paragraph we added "Such processes can also be assessed in fully coupled models, which enable a full range of interactions and feedbacks."

L320: Are the 1-year experiments suggested for atmosphere-only models, coupled models, or both?

Response: As stated in section 3, it is likely to be atmosphere-only with prescribed SSTs and sea ice for this 1-year experiment. To be clear for this manuscript, we have clarified this to specify atmosphere-only simulations, but it is true that it could be expanded to coupled simulations in the future.

Fig. 2 is one of the most important parts of the paper, yet I find it to be rather inefficient in presenting the suggested simulations. For example, the figure tends to be overloaded with text, the tiers of simulations are not given, and the link to table 2 could be made stronger.

Response: We will remove the superfluous text from the figure to make it clearer, and better label it to relate more strongly to Table 2.
Fig. 2 has been greatly simplified, with Tier and experiment names referring to Table 2 but other text removed.

L391: What is meant with "... as well as some comparison to HighResMIP1." This almost seems to say that previous participation of a model in HighResMIP1 is required, which I assume is not the case, however.

Response: This is not meant to suggest that previous participation is required. It is just meant to articulate that there is some similarity in the Tier 1 simulations between HighResMIP1 and HighResMIP2. We simply remove the sentence.

L415: "enable new science" This does not say much in my view, and is one of many examples to streamline the paper.

Response: We have removed this part of the sentence as it is repetitive.

L448: What is the correct way to average aerosol properties over time? Is a simple time average of aerosol properties okay, or would one need according to the time-varying aerosol mass when, for example, computing the time average single scattering albedo?

Response: Modelling groups setting up their simulations will best know how to configure their own aerosol scheme, or to implement an aerosol climatology (as they decide is appropriate). As we note, HighResMIP2 wants to be less restrictive on such choices compared to HighResMIP1, but strongly encourages groups to fully document the methods and forcings they use.
We have added a sentence at L423 to answer this broader question of forcings and implementation.

Fig. 4: What is recommended in terms of interpolation of SSTs in time (e.g., should daily-mean SSTs be read in) and space (i.e., how should SSTs be interpolated to the model grid)?

Response: A dataset that is consistent with the Tier 1a forcing dataset will be provided.  It will be up to modelling groups to interpolate as they routinely do for AMIP-like simulations. In addition, it is requested that daily-mean SSTs be read in. We have added text near to L429 to this effect.

L446: "solar, ozone,"?

Response: Changed to: "For solar and ozone forcings, it is…"

L467: Does the possibility of different forcings across models not endanger the usefulness of HighResMIP2?

Response: It, potentially, limits the usefulness of HighResMIP2, but in our discussions with modelling groups we found that having some flexibility here increases participation.  It is our assumption, however, that based on the discussions most groups will stick with the recommendation.  Again, we want to emphasise that Tier 1 simulations are the entry card to HighResMIP2.
(we make no change to text, apart from the addition at L429)

Typos etc:

L231: The brackets around Chang et al. are wrongly placed.

Response: Corrected

**Referee 2 Claudia Tebaldi**

As Referee n.1 posted, I don't find any reason to argue with the authors rationale and motivations. The protocol is also very thoughtful, and I like the emphasis on idealized experiments. As Referee n.1, I do encourage some streamlining and shortening of the paper if possible.

I was surprised, I have to say, that the authors of a MIP whose first motivation is (I quote) discussed in the IPCC AR6 report, and can it contribute to and supplement the CMIP7 AR7 FastTrack science goals, particularly around future weather extremes, the pattern effect (interaction of forcings, feedbacks and natural variability) and tipping points?" managed writing an entire paper without a single mention of ScenarioMIP, not even when they mention SSP5-8.5 and the desire to adopt a different scenario this time around. Why the complete lack of connection to what, I would think, is one of the main sources of those projections that the MIP aims at better quantify and characterize? Given the lack of connection (let alone communication) it is not clear to me if the authors plan to use one of the new scenarios that we are developing in ScenarioMIP. If they do (which I would imagine would also be the to push forward the date when the handshake between historical and future simulations happens, to 2022), they may be interested in connecting to us, among the other MIPs they mention. If they do, they'll find out that the new scenarios won't be labelled "ssp-something-something" anymore, and we are going to have at least two scenarios that may be of interest. One that we call "high" and will replace SSP5-8.5 with the highest plausible emissions, given current trends and a plausible narrative that may reverse some of those and create warming in excess of what current trends may imply. This scenario is specifically being developed to support the continuing interest of the impacts community to explore a wide range of warming levels, and possible tipping points. This scenario could well check the box for HighResMIP2. We will also have a "medium" scenario, where current policies will be frozen for the remainder of the century. While we expect the "high" to reach around 7W/m2 by 2100 (but not be rich in SLCF like the old CMIP6 SSP3-7.0), the medium will come to a lower level than 7.0W/m2 by 2100, but still significantly higher than scenarios aligned with the Paris agreement targets. Anyway, if the authors are interested they could reach out to us. The scenarios will be available summer of 2025.

All this said, maybe the plan is different but, in that case, more details on how the future scenario is going to be chosen and implemented with a later starting time would be good to see.

We thank the reviewer for their time and the very useful comments and suggestions.

Response: We apologise for not reaching out to the ScenarioMIP community and making the links between the HighResMIP2 simulations and ScenarioMIP explicit. We did consult the CMIP7 panel about their thoughts on future scenarios, and we recognised that ScenarioMIP were still developing the scenarios for CMIP7, so we did not want to get into too much detail at this point, but rather give some broad thoughts on what scenario(s) would be most use here, particularly (our assumption) that modelling groups will be constrained to one or only a few ensemble members. Given the expense of the HighResMIP simulations, we do not anticipate large ensembles of simulations, and it is unclear how useful a few members of (a range of) scenarios would be, but we would welcome advice on this.

In revising the manuscript, we will make this clearer, and make better reference to ScenarioMIP, and include some of the discussion above about which scenarios may be most useful and why. In particular we will read with interest what the new CMIP7 "high" scenario entails and whether this should be our recommendation.

Note - the authors have exchanged emails with Claudia now to discuss further some of these issues. The ScenarioMIP manuscript is due to be submitted imminently, so we will refer to that once it is available.

We have added text at L423 with a little more discussion on choices of scenario, but noted that HighResMIP2 simulations will not be able to explore a wide range of scenarios, but rather to see what higher resolution enables in terms of simulating future extremes, impacts and feedbacks.
We also added references to CMIP6 ScenarioMIP at L339, noted CMIP7 ScenarioMIP at L349 and L494, and changed the name of the highres-future simulations to remove the ssp.

**Reviewer 3**

This manuscript proposes and describes a second phase of the High Resolution Model Intercomparison Project (HighResMIP2). The justification for this MIP is discussed in detail, and all the proposed experiments are introduced.

The manuscript is very well written and seems to have benefitted from extensive discussion among the interested groups. My comments are mostly minor.

**Major Comments**

1) One concern is the lack of parallel standard-resolution simulations. Given that the models are not expected to include ESM components, we do not expect them to produce the most accurate climate change projections. Thus, to my mind, the main purpose of HighResMIP2 should be to assess the impact of resolution on climate models in a relatively simple experimental framework. In the absence of a companion simulation this will be difficult to achieve.

I understand the authors feel the need to set phase 2 apart from phase 1, but nevertheless wonder whether this new design will be able to do justice to what could be considered the core objective of this exercise.

We thank the reviewer for their time and the very useful comments and suggestions.

Response: This question was one that we discussed extensively within the HighResMIP working group, and has no easy answer. We agree that the high resolution models in HighResMIP will lack many aspects of Earth System complexity, and this may influence the climate projections, so including a low resolution counterpart model could help with interpretation in this regard. However, there are now many groups (including storm-resolving as you mention later) who focus on higher resolution modelling and have little interest in using typical CMIP-resolution models. We do not want to exclude such groups from HighResMIP simply because they are unable or unwilling to provide low resolution counterpart simulations. We still expect many groups to run lower resolution models anyway, but with the whole CMIP archive, and HighResMIP1 as existing reference datasets for low resolution models, we would argue that producing more low resolution simulations should not be a requirement of HighResMIP2 participation.
We have noted this at L408

2) There is a lot of emphasis on AMIP-type experiments. I assume some of these are important for climate sensitivity metrics. Nevertheless, this kind of experiment with prescribed SSTs can lead to unrealistic surface heat fluxes, as indicated in several studies. This is particularly the case in the middle and high latitudes, where SSTs tend to be driven by atmospheric heat fluxes (O'Reilly et al. 2023; Kim et al. 2024) but can also be problematic in the tropics (Wang et al. 2005). It can be expected that the impact of "rigid" SSTs becomes more detrimental when one moves to higher spatial resolution, where vigorous air-sea interaction is expected to occur due to resolved atmospheric and oceanic fronts.

Furthermore, while satellite observations yield SST data at high spatial resolution, the temporal coverage may not be sufficient to provide realistic sub-daily forcing across the globe.

These issues should be at least discussed when introducing the experiments and their justification.

Response: With regard to SST temporal coverage - we will use daily mean SST forcing, not sub-daily forcing, as we agree that the datasets are not available for this (this is now noted at L442).
This certainly removes the diurnal cycle from the ocean forcing, and hence the models may miss aspects of sub-daily variability. However, we would contrast our experiment with the standard CMIP AMIP2 experiment which uses monthly, 1 degree SST data and suggest we still learn a lot about climate simulation when the models are constrained in this way.
We will include further discussion on the limitations of SST-forced simulations, but it remains the simplest experiment to configure and compare across models. We will add the references given to this discussion.
We have added the reviewers concerns about SST-driven simulations at L460.

Could the priorities of the experiments be shifted in favor of more coupled ocean-atmosphere experiments?

Response: We somewhat disagree that the emphasis is more strongly on the AMIP simulations. It is true that in CMIP6 HighResMIP many more groups submitted the AMIP simulations than coupled, primarily because they are much more straightforward to set up and run. Our Tier 1 (entry-card) simulation can be either the AMIP-style or the coupled simulation, so groups are free to choose to do only the coupled simulations.
The text in the manuscript makes somewhat of a plea for increasing the number of eddy-rich coupled simulations (given their extreme paucity currently), as this is a huge gap in our climate knowledge. But clearly such simulations are very challenging in many ways.

Given this, we have not made any changes to the text.

**Minor Comments**

1) l. 116: What kind of predictions improved? Seasonal, decadal?

Response: As with reviewer 1, we have modified this to "with implications for decadal variability and climate projections". HighResMIP1 simulations themselves were not part of e.g. CMIP6 decadal predictions, so we cannot talk about skill directly. It is true that the model resolutions referenced (and used for decadal hindcast predictions) are consistent with those used in HighResMIP1, and show improved skill, but we leave this out.

2) ll. 149-151: I do not follow the argument here. Why is the fact that HighResMIP1 used CMIP6 forcing "evidence that the details of the forcing … are not central"? How can you tell, if the CMIP7 forcing is not available and therefore no experiments have been performed with it yet?

Response: We apologise for the lack of clarity. What we are trying to say is that the differences we found in CMIP6 HighResMIP were due to model resolution. Hence we are suggesting that for CMIP7, indeed the forcing might change the model simulations, but there is little reason to assume that the delta(resolution) response would be any different. This is clearly our assumption, and we will make this clearer in the text.
We have deleted the sentence, as on reflection it does not add to our argument.

Also, in Table 1, "forcings used" in the rightmost column is a bit confusing, because these experiments have not been performed yet. Maybe "forcings to be used" would be better, or just "forcings".

Response: Agreed, we have changed this to forcings.

3) ll. 217-219: Is there reason to expect that eddy-resolving ocean simulations will be a game changer? The statement is repeated later, but I wonder how much support there is in the literature.

Response: We think there is indeed evidence of this, from the CESM-HR simulations (citations of Chang et al., Yeager et al. and others have been added here) referenced in the manuscript. There are very few existing multi-centennial eddy-rich coupled simulations, but those that do exist (see references above) are producing a huge amount of new insights.

It is interesting that the authors do not mention cloud-system resolving capabilities. There are certainly expectations that turning off cumulus parameterization schemes would eliminate one of the key weaknesses of climate models. Some of the models listed in Table 1 seem to be at least close to that resolution.

Response: We did not address this in detail, as we are not sure how many such models will be able to complete full 1980-2022 simulations as part of Tier 1 (though we do note the potential to build much stronger links with e.g. DYAMOND and other communities). However, this is exactly what the 1-year Tier 5 simulations are designed for, to enable better comparisons of models with different physics/parameterisation settings, using the same forcings. We will make this clearer in the text.
As noted with Reviewer 1, we leave the model choices, such as how to represent atmospheric convection, to the modelling groups themselves, as there are a range of views and implementations.
We have added at L197 that storm-resolving models would of course be welcome to complete HighResMIP2 simulations.
L246 (as for Reviewer 1) we note that physics choices are left to the modelling groups.
L266 we note some examples of how resolutions of 10km and beyond might give new insights.

4) ll. 251-261: The limitations of atmosphere-only experiments are not discussed (see major comment 2). Particularly in the midlatitudes, where the atmosphere strongly influences the ocean, a simulation with fixed SST may lead to unrealistic surface heat fluxes.

Response: We recognise that AMIP-style simulations are not ideal, but they are the standard type of simulation for CMIP, are straightforward to set up, and allow the models to be compared with each other easily. We do recognise that in removing interactive coupling we strongly limit key processes, and we will make this clearer in the text.
As for your major comment, L460 notes these limitations.

5) ll. 262-264: The one-year simulations seem to overlap with the DYAMOND project. What is the rationale for repeating these experiments? In the introduction you mention that this project lacks "basic characterization". I assume you mostly mean that the climate sensitivity is not necessarily known for the DYAMOND models? A bit more explanation would be helpful.

Response: We indeed want to coordinate with DYAMOND and certainly not repeat what that project plans, choosing the same year for simulation is deliberate in this regard. However we think there is still a role for HighResMIP to compare models

with different resolutions, from "CMIP-type" models to km-scale, and testing metrics (such as climate sensitivity) calculated from 1 year, to longer AMIP-style simulation, and then (if available) coupled simulations. We will expand on the text to explain better.
L559 added sentence to make clear that this is a bridge/link between communities not a repeat.

6) ll. 338-340: Since many participating models will probably lack ESM components (as stated by the authors), I see the role of HighResMIP2 more as idealized experiments to probe the sensitivity to model resolution (see major comment 1). As such, I feel it would be more interesting to use a high-emission scenario in order to obtain a robust response of the climate system.

Incidentally, it would be of interest to check which of the CMIP6 scenarios followed most closely the observed greenhouse gas concentrations from 2015-present.

Later on, the authors cite the study of Meinshausen et al. 2024 to support the notion that the CMIP6 high-resolution scenarios are unrealistic. I did not read that paper thoroughly, but a quick look did not turn up any statement or graphic to that effect. Instead, in the abstract, I found the following statement: "a higher-emission category that is approximately in line with "current policies" (as expressed by 2023)".

I am actually interested in this topic and would appreciate if the authors could point me to the relevant passage in the paper, or to a different paper that discusses this.

Additional references may also be helpful for the present manuscript.

Response: It is true that HighResMIP simulations will lack ESM complexity, and hence will have limitations in their future projections - just as lower resolution ESM models have limitations in terms of climate variability, extremes and other aspects. So all simulations are adding to our understanding of ranges of future projections and likely risks.

We have added the 4xCO2 simulation with the explicit goal of providing an idealised warming scenario as used elsewhere in CMIP (though with a different baseline here). Removing the restrictions recommended in HighResMIP1 in terms of aerosol forcing and the like, as well as encouraging models configured explicitly for high resolution, running to 2100 and including warming-level experiments, have the potential to make the HighResMIP2 simulations much more relevant for future projections and impacts, able to be contrasted with those from standard CMIP7 simulations. In this regard, using a "likely" future scenario, not just an extreme one (which many recent articles have strongly criticised), makes sense. As we noted, it was the CMIP7 panel who recommended the scenario we suggest, and we have

read the ScenarioMIP manuscript that reviewer 2 mentioned, which confirms our choices.

With regard to which scenario the world is currently following, we agree that the Meinshausen et al. 2024 paper did not say exactly what we inferred. We have replaced this reference (L490), and also include several more references as below which do suggest not using SSP5-8.5. As in our response to Reviewer 2, we have read with interest the new CMIP7 scenarios and refer to that in the revised manuscript.

Shiogama, H., Fujimori, S., Hasegawa, T. et al. Important distinctiveness of SSP3–7.0 for use in impact assessments. Nat. Clim. Chang. 13, 1276–1278 (2023). https://doi.org/10.1038/s41558-023-01883-2

Climate Action Tracker: Warming Projections Global Update—November 2022 https://go.nature.com/47jNhP4 (Climate Analytics & NewClimate Institute, 2022).

Hausfather, Z. & Peters, G. P, 2020: Emissions – the 'business as usual' story is misleading. Nature 577, 618–620 (2020), https://doi.org/10.1038/d41586-020-00177-3

Thank you for the additional references.

7) Figure 2: What does "pxxkpat" stand for?

Response: Apologies, we will make this clearer. It stands for "warming level (xx) k, patterned" with k=2,3,4 K etc. This is clarified in Table 2, and the text removed from Fig. 2 (at the request of Reviewer 1).

8) Table 2: CMIP6 used to have three tiers for experiments but HighRESMIP2 proposes five. Is this a new feature of CMIP7 or particular to this MIP? Seeing how Tier 3 experiments in CMIP6 tend to have relatively little uptake, I am wondering how much participation you expect for the Tier 4 and 5 experiments.

Response: We are not aware that there is a maximum number of Tiers, so having five Tiers is our own choice based on extensive discussions with modelling groups. We agree that typically the uptake in the higher Tier experiments is less. However, Tier 5 is the 1 year experiment (and hence relatively cheaper, potentially interesting a different community), and Tier 4 will be particularly of interest to groups interested in warming levels and the impact on extremes (we have been actively engaged with such people who wanted these experiments included).

9) Figure 4: Why is the ESA CCI SST trend (Fig. 4c) smoother than the one in HighResMIP (Fig. 4b)? Is there any smoothing or regridding applied?

Response: We are not completely sure. However, it is likely a consequence of the way that the HighResMIP1 SST forcing was calculated. Since HighResMIP1 started in

1950, we needed to derive a daily, 0.25 degree grid dataset, which clearly means that assumptions had to be made. For HighResMIP2, starting the AMIP simulation in 1980 means we can use satellite-era data throughout, and this likely removes the noise. We have noted this in the text.

10) Figure 5: Is this from a single model or a multi-model average? Which models are used?

Response: This is a single model, and is only for illustration of what we might be able to say from HighResMIP2 (there was no formal 4xCO2 simulation in HighResMIP1). We will try to make it clearer that it is only meant to be an illustration from one model – we have noted that is it one model in the caption.

**References**

Kim, W. M., Y. Ruprich-Robert, Y., A. Zhao, S. Yeager, and J. Robson, 2024: North Atlantic Response to Observed North Atlantic Oscillation Surface Heat Flux in Three Climate Models, *J. Climate*, **37**, 1777–1796, https://doi.org/10.1175/JCLI-D-23-0301.1.
O'Reilly, C. H., et al., 2023: Challenges with interpreting the impact of Atlantic Multidecadal Variability using SST-restoring experiments. *npj Clim Atmos Sci*, **6**, 14 (2023). https://doi.org/10.1038/s41612-023-00335-0
Wang, B., Q. Ding, X. Fu, I.-S. Kang, K. Jin, J. Shukla, and F. Doblas-Reyes, 2005: Fundamental challenge in simulation and prediction of summer monsoon rainfall, *Geophys. Res. Lett.*, **32**, L15711, doi:10.1029/2005GL022734.

---

## Author Response (AR2)

HighResMIP2 reviewer comments on revised version

**Report 1**

Thank you for addressing my suggestion and connecting explicitly to the new scenario protocol.
As far as I am concerned the paper should be accepted. I just have a small suggestion/correction related to the mention of the Medium scenario. Right now you say: "we propose to recommend their Medium future scenario that will produce an additional radiative forcing, at the end-of-century, of around 5.3 W m-2". As you know, in our paper we present those scenario trajectories as illustrative, based on estimates from a simple climate model (FaIR to be specific). We really do not know what the RF for these scenarios will be, especially this time around, as we recommend they be run in emission-driven mode. Could I suggest to slightly rephrase and say: "we propose to recommend their Medium future scenario, estimated to produce a radiative forcing, at the end-of-century, around or possibly slightly above 5 W m-2" ? Feel free to reword, but I would like to communicate some fuzziness here, and 5.3 seems awfully precise. I also deleted "additional" as it is not clear to me what that would be additional to, but again feel free to push back on that deletion if you have strong opinions, as long as it is clear that that number (5Wm-2) is RF compared to 0 (pre-Industrial). Thank you again. At this time the ScenarioMIP paper is still in GMD limbo, will let you know as soon as it is fully citable. Claudia

**Response:**
Many thanks Claudia.
We have made this exact change in the text around L477.